# Dual client binding sites in the ATP-independent chaperone SurA

Bob Schiffrin [1,5], Joel A. Crossley [1,5], Martin Walko [1,2], Jonathan M. Machin[1], G. Nasir Khan[1], Iain W. Manfield [1], Andrew J. Wilson [2,4], David J. Brockwell [1], Tomas Fessl[3], Antonio N. Calabrese [1], Sheena E. Radford [1] ✉ & Anastasia Zhuravleva [1] ✉

The ATP-independent chaperone SurA protects unfolded outer membrane proteins (OMPs) from aggregation in the periplasm of Gram-negative bacteria, and delivers them to the β-barrel assembly machinery (BAM) for folding into the outer membrane (OM). Precisely how SurA recognises and binds its different OMP clients remains unclear. *Escherichia coli* SurA comprises three domains: a core and two PPIase domains (P1 and P2). Here, by combining methyl-TROSY NMR, single-molecule Förster resonance energy transfer (smFRET), and bioinformatics analyses we show that SurA client binding is mediated by two binding hotspots in the core and P1 domains. These interactions are driven by aromatic-rich motifs in the client proteins, leading to SurA core/P1 domain rearrangements and expansion of clients from collapsed, non-native states. We demonstrate that the core domain is key to OMP expansion by SurA, and uncover a role for SurA PPIase domains in limiting the extent of expansion. The results reveal insights into SurA-OMP recognition and the mechanism of activation for an ATP-independent chaperone, and suggest a route to targeting the functions of a chaperone key to bacterial virulence and OM integrity.

The outer membrane (OM) of Gram-negative bacteria is key to cell survival and virulence, providing structural integrity and presenting a barrier to antibiotics, bile salts and detergents[1,2]. The OM bilayer is densely packed with OM proteins (OMPs)[3–5], that form cylindrical β-barrel structures of varying size (8–26 strands in *E. coli*)[6–8]. These proteins perform a variety of functions, including nutrient acquisition, efflux of toxic molecules, and OM assembly, and play critical roles in pathogenesis[8]. OMPs have a complex biogenesis pathway: they are synthesised on cytoplasmic ribosomes and bind to the chaperones Trigger Factor and SecB, before their Sec mediated secretion into the periplasm[9]. Therein, they are maintained in a soluble and folding-competent state by periplasmic chaperones (e.g., SurA, Skp, FkpA, and

Spy)[10,11], before being folded into the OM by the essential OMP complex known as the β-barrel assembly machinery (BAM)[12–14]. In *E. coli*, BAM is a ~203 kDa heteropentameric complex, consisting of the conserved subunit BamA, itself a 16-stranded OMP, and four lipoproteins (BamB-E)[15–18].

Current evidence suggests that OMPs fold in vivo by sequential formation of β-strands that is templated by the N-terminal β-strand of BamA's β-barrel (β1) making a β-augmentation interaction with the C-terminal β-strand of the incoming OMP substrate[7,19–22]. The C-terminal strand in OMPs typically contains a β-signal sequence [ζxGxx[Ω/Φ]x[Ω/Φ] where ζ is a polar residue; Ω is an aromatic residue; Φ is a hydrophobic residue and x is any residue. The β-signal

[1]Astbury Centre for Structural Molecular Biology, School of Molecular and Cellular Biology, Faculty of Biological Sciences, University of Leeds, Leeds, UK. [2]Astbury Centre for Structural Molecular Biology, School of Chemistry, University of Leeds, Leeds, UK. [3]Faculty of Science, University of South Bohemia, České Budějovice, Czech Republic. [4]Present address: School of Chemistry, University of Birmingham, Edgbaston, Birmingham, UK. [5]These authors contributed equally: Bob Schiffrin, Joel A. Crossley. ✉e-mail: s.e.radford@leeds.ac.uk; a.zhuravleva@leeds.ac.uk

sequence, and in particular its final residue, plays a pivotal role in OMP assembly in vivo[23,24] and in vitro[24–26]. Consistent with this model, a recently discovered antibiotic, darobactin, acts as a potent antibacterial by mimicking and inhibiting the OMP: BamA β-augmentation interaction at this site[27–29]. This model of BAM-catalysed OMP assembly suggests that chaperones need to deliver unfolded OMPs to BAM for their folding in a vectorial manner, commencing with their C-terminal β-strand.

SurA is thought to be the major OMP chaperone in the periplasm and the primary route for OMP delivery to BAM[30,31]. Deletion of SurA leads to loss of OM integrity, upregulation of the σ$^E$ envelope stress response and increased permeability of the OM to large antibiotics, such as vancomycin and rifampicin[30,32–36]. SurA is also involved in the biogenesis of virulence factors, including adhesins, pili, and autotransporters[37–46], and deletion of SurA restores antibiotic sensitivity in a multidrug resistant *Pseudomonas aeruginosa* strain[47]. In *E. coli*, SurA contains three domains: a core comprising the N-terminal and C-terminal regions, and two tandem peptidyl prolyl isomerase (PPIase) domains (P1 and P2), only the latter of which is enzymatically functional (Fig. 1a)[48]. The isolated core domain has chaperone activity[49], and multiple lines of evidence point to functional roles for the P1 and/or P2 domains in facilitating the prevention of OMP aggregation and the delivery of OMPs to BAM[35,41,49–52]. Dynamic domain rearrangements between the core and PPIase domains have been proposed to play a role in SurA activity (Fig. 1a, b), although how this disfavours aggregation and promotes delivery to BAM is unclear[50,53–56].

How SurA achieves its specificity for OMP sequences is also unknown, with previous studies indicating a preference for Ar-X-Ar and Ar-Ar containing sequences[57] (where Ar is an aromatic residue and X is any amino acid), or aromatic-enriched sequences containing an Ar-X-Ar-X-Pro motif[58]. Accordingly, the peptide WEYIPNV was shown to

bind SurA with μM affinity ($K_d$ 1–14 μM)[58], with X-ray crystallography showing that this peptide binds to the P1 domain[56,59]. This led to a model for SurA-client recognition in which the P1 domain confers SurA's specificity for OMP sequences, with the core domain acting as a more general chaperone[35,59]. Structures of a second aromatic-rich peptide (NFTLKFWDIFRK) bound to the P1 domain (PDB: 2PV2), and to a SurA construct lacking the P2 domain, named SurA-ΔP2 (PDB: 2PV3), support the view that the P1 and/or core domains are important players in SurA-client recognition[59]. It remains unclear, however, whether SurA-client interactions involve a diffuse binding interface between core and P1, or if binding involves specific binding sites, and how SurA inter-domain re-arrangements, that are known to occur upon client binding[56], mediate its chaperone function.

Here we address these questions using methyl-TROSY NMR, smFRET, binding assays using different OMPs and OMP-derived peptides, and bioinformatic analysis of conservation across SurA sequences. The results reveal that SurA has two conserved client binding hotspots that mediate affinity for unfolded OMP substrates, one in the core domain and one in the P1 domain, and add a distinct dimension to previous models suggesting that SurA binds its clients via a large and diffuse binding interface[54,56]. The discovery of the two binding hotspots provides an opportunity to target these sites with small molecules to attenuate virulence and increase bacterial susceptibility to antibiotics by inhibiting a chaperone key to biogenesis of the OM.

## Results

### SurA domains tune the conformational properties of a bound OMP client

We began our investigations into the role of the different SurA domains in OMP binding by examining the conservation of PPIase domains in SurA homologues across bacterial species. Using

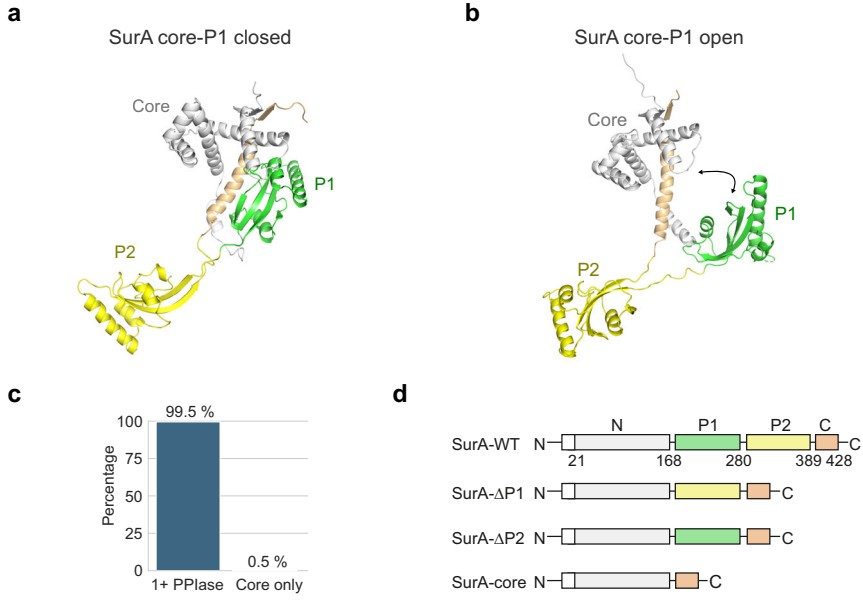

**Fig. 1 | Structure, conformations and domain architecture of *E. coli* SurA.**
**a** Crystal structure of wild-type (WT) *E. coli* SurA (PDB: 1M5Y[48]) with missing residues added using MODELLER[56,119]. In this structure the P1 domain is bound to the core domain (named here, SurA core-P1 closed). Regions are coloured grey (N-terminal region of the core domain), green (P1), yellow (P2) and orange (C-terminal region of the core domain). This colour scheme for SurA domains is used throughout. **b** Model of WT SurA in which the P1 domain is extended away from the core domain (named here SurA core-P1 open). The model was built using

MODELLER[56,119] and the crystal structures of full-length SurA (PDB: 1M5Y[48]) and SurA-ΔP2 (PDB: 2PV3[59]). **c** Bar chart showing the percentage of SurA homologues from the InterPro family IPRO15391 (14,422 homologues) which contain at least one PPIase domain (1+PPIase) or are core domain only (Core only) homologues.
**d** Domain architecture of *E. coli* SurA-WT and the SurA domain deletion variants used in this study. The signal sequence was not present in the constructs used here, but the numbering used reflects the gene numbering and includes the signal peptide (residues 1–22) (white). Source data are provided as a Source Data file.

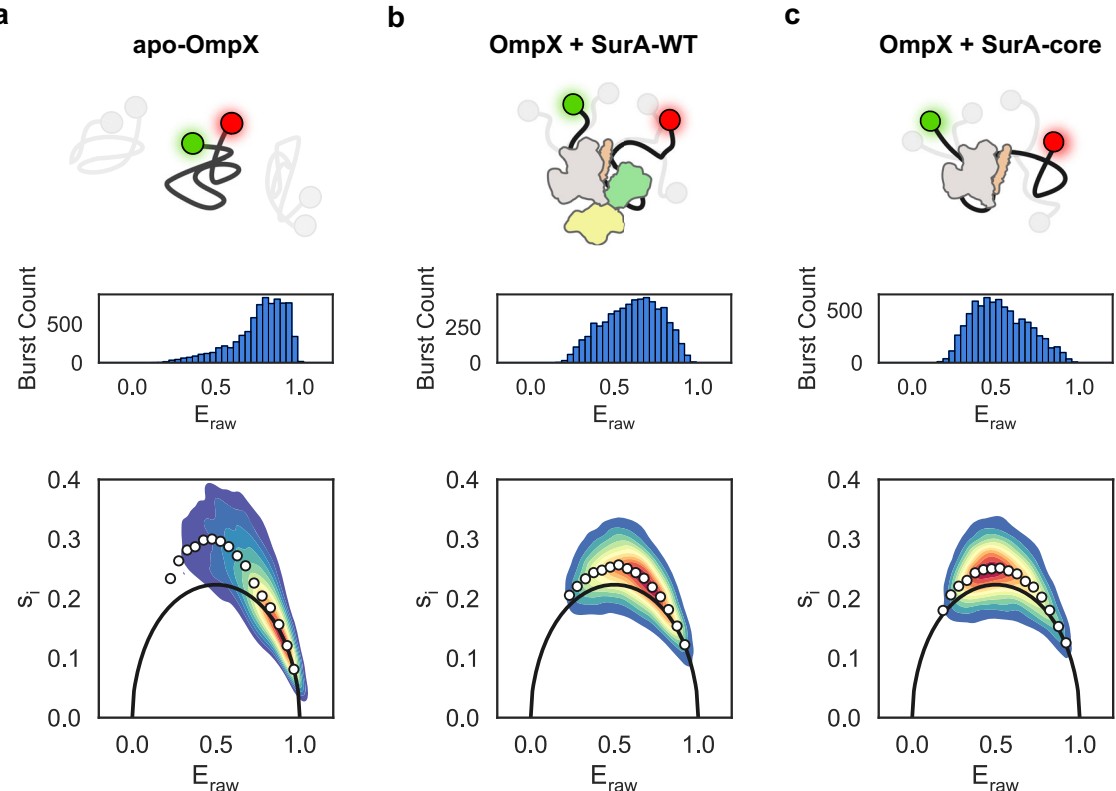

**Fig. 2 | smFRET captures the conformational dynamics of unfolded OmpX in the presence or absence of SurA variants. a** Schematic showing OmpX (black line) with donor and acceptor fluorophores on the N- and C-termini (green and red) with example alternate conformers shaded in grey (top), $E_{raw}$ histogram (middle) and BVA analysis (lower) for apo-OmpX. The solid black line represents the expected standard deviation for a static (ms timescale) FRET population. The white circles show the mean $s_i$ for bin widths of 0.05 of $E_{raw}$. **b** OmpX + SurA-WT and (**c**) OmpX + SurA-core with panels as in (**a**). The schematic of SurA is coloured according to Fig. 1. Source data are provided as a Source Data file.

AlphaFold2 (Methods)[60,61], we determined the number and taxonomy of organisms containing SurA sequences that contain only a core domain, versus those that contain one or more PPIase domains. The analysis revealed that >99% of SurA homologues contain at least one PPIase domain (99.5% of 14,244 homologues) (Fig. 1c), with the few examples of sequences which lack PPIase domains largely restricted to bacteria specialised to extreme environments (Source Data file). This suggests that the presence of at least one PPIase domain is functionally important for bacterial survival and/or virulence.

Previous small-angle neutron scattering (SANS) and smFRET experiments have shown that binding to SurA expands its (unfolded) OMP clients[54,62,63]. However, the roles of the different SurA domains in expansion were unknown. To address this question, we used pulsed interleaved excitation smFRET (Supplementary Methods) to monitor the effects of binding SurA-WT, or a construct lacking both PPIase domains (SurA-core) (Fig. 1d), on the conformation and dynamics of the model OMP, OmpX[56,64,65]. Building on recent work[63], we labelled the N- and C-termini of OmpX with a FRET donor and acceptor dye pair (Methods), and used smFRET to monitor OmpX conformations under different conditions. Fluorescence anisotropy experiments indicated that both dyes at both termini have sufficient rotational freedom to allow changes in FRET values to be assigned to OmpX end-to-end distances (Supplementary Table 1, Supplementary Methods).

Unfolded OmpX diluted from high concentrations of denaturant into low (0.1 M) urea (apo-OmpX) yielded a broad raw FRET distribution ($E_{raw}$) with an maximum at ~0.85, suggesting a distribution of conformational species ranging from collapsed to more expanded species in rapid equilibrium (Fig. 2a, upper two panels). Non-random-coil chain behaviour of unfolded OMPs in low amounts of denaturant is well-established in the field[54,63,66], as might be expected for an unfolded

protein poised to fold once it encounters a bilayer. By contrast, OmpX fully unfolded in 4 M urea resulted in a narrower distribution with a significantly lower $E_{raw}$ maximum (~0.45) (Supplementary Fig. 1d), consistent with a more highly unfolded ensemble, as reported previously[63,66,67].

When apo-OmpX is bound to SurA-WT (in 0.1 M urea) the smFRET distribution broadens and its maximum is shifted to a lower $E_{max}$ value (~0.7) compared with that of apo-OmpX, consistent with a shift in equilibrium to favour more expanded conformations. Hence, binding to the chaperone results in a net expansion of the collapsed chain, as previously reported[63] (Fig. 2b, upper two panels). Remarkably, the addition of SurA-core shifts the $E_{raw}$ values of OmpX to even lower $E_{max}$ values compared to the protein alone and bound to SurA-WT ($E_{raw}$ maxima of ~0.45), with a concomitant depletion of species with higher apparent $E_{raw}$ values (Fig. 2c, upper two panels). This suggests that the P1 and P2 domains modulate the conformations and dynamics of SurA-bound OMP clients. Both SurA-WT and SurA-core have been shown to bind to unfolded OMPs with similar affinity (~μM) and with Hill coefficients >1[49,50], suggesting that multiple copies of SurA variants could be bound to each unfolded OmpX chain. One contribution to the greater expansion observed for SurA-core could be additional SurA-core molecules bound to OmpX compared with SurA-WT, although microscale thermophoresis (MST) experiments indicated similar Hill coefficients for the binding of OmpX to both variants, consistent with previous results[50] (Supplementary Fig. 2).

We next investigated the timescales of dynamics of the OmpX chain of the different samples apo-OmpX (in 0.1 M and 4 M urea) and bound to SurA-WT or SurA-core (in 0.1 M urea). Using Burst Variance Analysis (BVA) and corrected FRET efficiency vs normalised fluorescence lifetime we probed for dynamics on ms and faster timescales[68,69]

(Methods). Dynamic exchange between species with different E values are observed on both timescales for unfolded apo-OmpX in 0.1 M urea, as well as OmpX bound to SurA-WT and SurA-core (Fig. 2a–c, lower panels, Supplementary Fig. 1). This is consistent with the behaviour expected for a collapsed unfolded chain where dynamics are limited by inter residue interactions in the unfolded-state ensemble[66]. By contrast, ms dynamics are not observed in OmpX in 4 M urea in the BVA analysis (Supplementary Fig. 1d) while faster dynamics are observed as evidenced by the lifetime analysis (Supplementary Fig. 1d), consistent with the behaviour expected for a fully unfolded polymer[66].

Collectively, the results show that binding to the SurA core domain alone is sufficient to cause a net expansion of unfolded OmpX. In addition, they suggest a role for the PPIase domains in limiting the extent of expansion of the bound client compared to SurA-core, which causes the greatest degree of expansion and lacks these domains.

## The SurA core and P1 domains are involved in client binding

Previous smFRET experiments have shown that client binding alters the interactions between the P1 and core domains of SurA[56], although data are currently lacking on precisely where, and how, the different SurA domains interact with their clients and the residues that determine their molecular recognition. To map the client binding regions on SurA in more detail we used site-specific methyl labelling and TROSY NMR. We specifically labelled the δ1 carbon atoms of isoleucine (29 residues) and ε carbon atoms of methionine (14 residues) with $^{13}$C, providing residue-specific probes that span all three domains of SurA (Supplementary Fig. 3a–c)[70,71]. Of the resulting 43 peaks in the spectrum of apo-SurA (i.e., without client), 33 were not overlapped or broadened, and could be assigned unambiguously using mutation (see Methods), providing probes for all three domains (18 in the core, 7 in P1, and 8 in P2). While the majority of peaks have high intensity, three non-overlapped peaks were broadened and of low intensity (M46, I217 and M400) (Supplementary Fig. 3d, e). These three residues are located at, or close to, the core-P1 interface in the crystal structure (Supplementary Fig. 3f, g), suggesting that core-P1 conformational rearrangements occur on an intermediate NMR timescale (~µs-ms)[72]. These conformational rearrangements are likely due to interconversions between the core-P1 closed and open states (Fig. 1a, b), as observed using smFRET[56] and SANS[53]. Analysis of chemical shift changes for the SurA domain deletion variants SurA-ΔP1 (that lacks the P1 domain), SurA-ΔP2 (that lacks the P2 domain) and SurA-core (that lacks both PPIase domains) (Fig. 1d), confirmed that the P1 domain indeed makes extensive interactions with the core, while P2 is more remote (Supplementary Figs. 4, 5).

We next analysed the effect of adding unfolded OmpX on the chemical shift and linewidths of SurA-WT $^{13}$CH$_3$ resonances using NMR (Fig. 3a–e, Supplementary Fig. 6). Both of these types of peak changes (shifts and/or broadening), can report on binding or conformational changes caused by binding at the location of the corresponding $^{13}$CH$_3$ probe[73,74]. To ensure a substantial fraction of bound SurA (chosen as ~60% given a K$_d$ of ~1 µM[56,63]) we employed 5 µM concentrations of both SurA and unfolded OmpX. This ensured sufficient binding, and good signal to noise within a few hours data acquisition (Methods), while maintaining an OMP concentration low enough to minimise aggregation[75]. Reductions in the peak intensity of methyl-TROSY resonances of SurA-WT were observed upon the addition of unfolded OmpX, whereas only small chemical shift perturbations (CSPs) were detected (Fig. 3c, Supplementary Fig. 6). Such behaviour (changes in intensity, but only small CSPs) has been observed previously for interactions between other chaperones and their unfolded substrates[76–78], suggesting heterogeneous binding interactions and/or intermediate exchange dynamics. Notably, the intensities of most peaks corresponding to residues in the SurA core domain were affected, consistent with this domain being a major location of OMP interactions[35,49,56] (Fig. 3c). Similarly, 6 out of 7 peaks corresponding to

residues in the P1 domain were reduced in intensity by ~50% upon OmpX binding, with one peak (M231) severely affected. By contrast, the peaks representing residues in the P2 domain were much less affected, with small (10–40%) reductions in intensity for all resonances, likely due to the slower molecular tumbling of the chaperone in its client-bound state[79]. We mapped the Z-scores (Methods)[80] of the SurA-WT peak intensity changes in the presence of OmpX (Fig. 3d) onto a model of the chaperone in a core-P1 open state (Fig. 3e). This analysis showed that regions in the core and P1 domains directly interact with OmpX, while P2 is largely uninvolved in OmpX binding, consistent with previous cross-linking mass spectrometry (MS) and hydrogen exchange MS (HDX-MS) data[56].

Next, to examine the effect of client size and sequence on SurA interactions, we studied the interactions between SurA and different OMPs: tOmpA (19 kDa), OmpF (37 kDa), and tBamA, the transmembrane domain of BamA (43 kDa) (Supplementary Fig. 7a). These clients bind SurA-WT with K$_d$ 3.7 ± 1.0 µM, K$_i$ 5.2 ± 1.7 µM, and K$_d$ 1.4 ± 0.1 µM for tOmpA[50], OmpF[81] and tBamA (Supplementary Fig. 7b), respectively (the inhibition constant (K$_i$) for binding of OmpF to SurA was obtained from phage-based ELISA assays[81]). In each case, the perturbation patterns in NMR resonances were similar to those observed when SurA binds OmpX, with key interactions involving the core and P1 binding sites, and P2 largely uninvolved (Supplementary Fig. 8).

## OMP binding hotspots in the core and P1 domains of SurA

While the NMR data presented above suggest that OmpX engages in widespread interactions involving the SurA core and P1 domains, this raises the question of which interaction sites on SurA are key for OmpX recognition and binding affinity. To identify whether, and how, different regions of SurA interact with different residues in the sequence of OmpX, we examined the effect of binding of six short (15 residue) peptides on the NMR spectrum of methyl-labelled SurA-WT. These peptides correspond to regions that form β-strands in native OmpX and also contain Ar-X-Ar or Ar-Ar motifs that have been implicated in SurA binding[57,58,81] (Fig. 3a, b, Supplementary Table 2). While all OmpX-derived peptides were found to bind to SurA (Supplementary Figs. 9,10), their affinities (K$_d$s > ~100 µM, Supplementary Fig. 11) are more than 100-times weaker than that of full-length OmpX (K$_d$ of ~1 µM[56,63]), implying avidity effects when SurA binds its full length OMP clients.

The addition of the OmpX-derived peptides (200 µM) to SurA (5 µM) also results in substantial changes (intensity reductions and/or CSPs) in the core and/or P1 domains, but with the P2 domain mostly unperturbed (Fig. 3f, g, Supplementary Figs. 9, 10), consistent with the results obtained using full-length OmpX (Fig. 3c, d). However, by contrast to the results for full-length OmpX, peptide binding to the core domain results in the most substantial changes to residues located between the two lobes of the N-domain (residues I91, M109, M114, I132, and M136), with other core domain residues being less, or not affected (Fig. 3h, Supplementary Figs. 9, 10). In the crystal structure of SurA-WT a helix from the core-P1 linker region of a neighbouring molecule in the crystal lattice was found at this location, suggestive of a putative, functional protein-protein interaction site, consistent with our finding that this region is directly involved in client binding (Fig. 3i, Supplementary Fig. 12a). In addition to perturbations to residues in the core domain, the binding of three of the five Ar-X-Ar containing OmpX peptides (QMN (corresponding to the second β-strand of OmpX, named β2), KHD (β6) and SVD (β8), and the Ar-Ar containing peptide NKN (β4)), also affect the peak for M231 in the P1 domain (Fig. 3f–h, Supplementary Figs. 9, 10). This residue is located in the region where the peptide WEYIPNV binds in the crystal structure of the WEYIPNV-P1 complex (Fig. 3i, Supplementary Fig. 12b)[59]. Previous results also showed that diazirine-labelled OmpX forms cross-links to a residue (E248) located close to this site[56]. In the NMR spectrum of SurA-WT bound to WEYIPNV (Supplementary Fig. 13a–h) we also observed

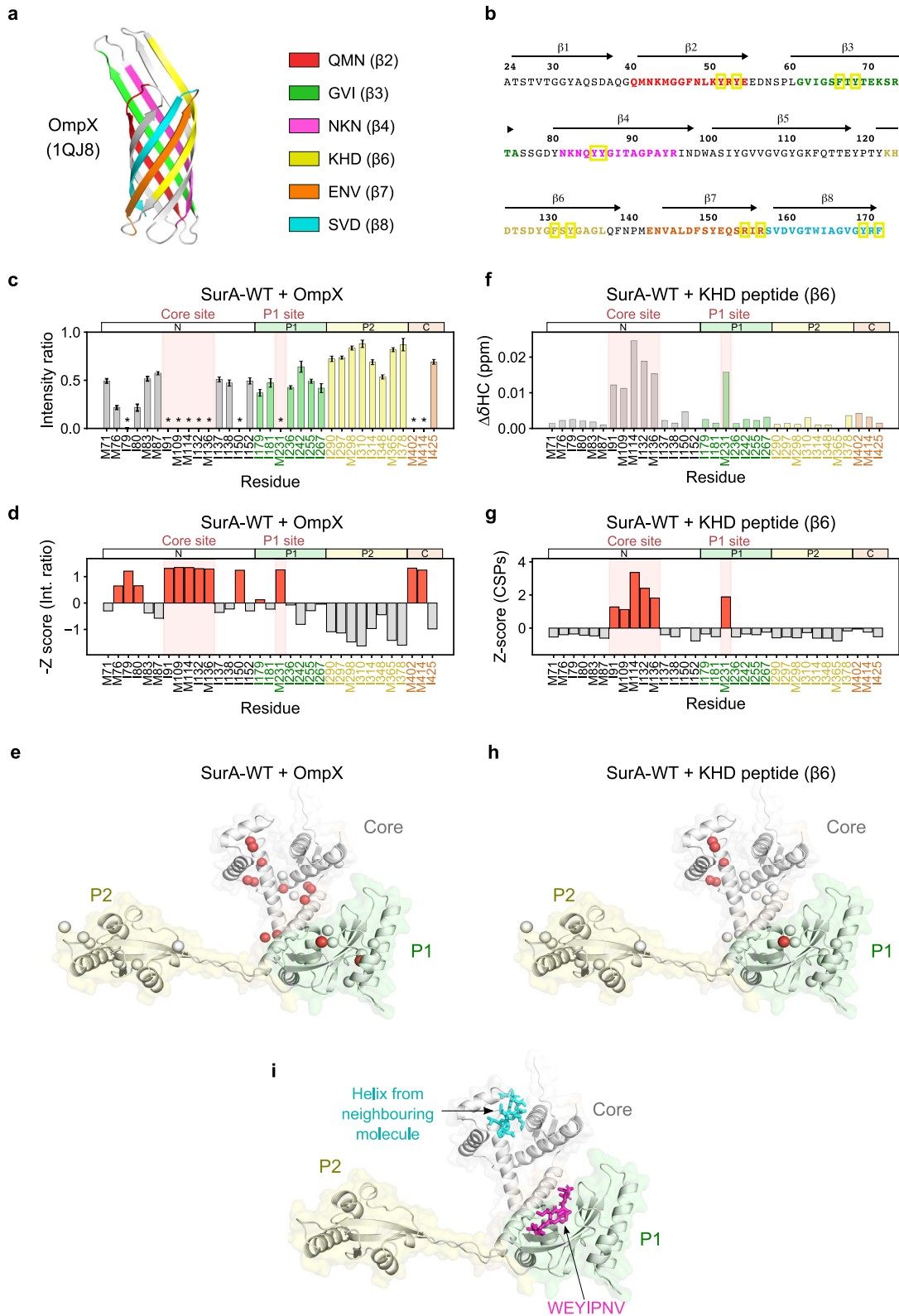

severe broadening of the resonance for M231, indicating that this peak is a reporter of direct interaction at this site. Interestingly, patches of high conservation are observed at both the core and P1 domain binding sites determined here (Supplementary Fig. 12c, d), consistent with these interaction hotspots being important for SurA function.

We also noted that the magnitude of changes in peak intensity and CSPs in the SurA core and P1 peptide binding sites varies significantly for the different OmpX peptides. For example, binding to the peptide GVI (OmpX β3) results in significant perturbations in resonances arising from core residues, but only subtle perturbations to those in the P1 binding site (Supplementary Figs. 9, 10). By contrast, binding to NKN (OmpX β4) most severely affects residues in the P1 site and results in smaller perturbations in the core domain (Supplementary Figs. 9, 10). The different peptide sequences, therefore, have different kinetic

**Fig. 3 | Dual binding hotspots in SurA mediate affinity for OMP substrates. a** OmpX-derived peptides analysed here mapped onto the crystal structure of natively folded OmpX (PDB: 1QJ8[64]). **b** OmpX sequence highlighting peptides analysed, coloured as in (**a**). Ar-X-Ar and Ar-Ar motifs are highlighted with yellow boxes. **c** Raw intensity change data and (**d**) Z-score plots of peak intensity ratios for SurA-WT binding to OmpX. Peaks which are broadened below the noise in the intensity ratio plots are indicated by an asterisk. For these peaks, the noise value for the bound spectra was used in the calculation of Z-scores. Signal to noise ratios were used for the calculation of errors in peak intensities (Methods). **e** Residues showing significant Z-scores for SurA-WT binding to OmpX mapped onto a model of SurA in a core-P1 open conformation. The ¹³C- labelled δ1 and ε carbon atoms of Ile and Met residues, respectively, are shown as spheres coloured by Z-score (red

for Z-scores > 0 and white for Z-scores <= 0). **f** Raw CSP data and (**g**) Z-score plots of CSPs for SurA-WT binding to the OmpX-derived KHD (β6) peptide, comprising a 15-residue sequence that forms β-strand 6 in the OmpX native state. **h** Residues showing different CSP Z-scores for SurA-WT binding to the KHD (β6) peptide mapped onto a SurA core-P1-open model (coloured as in (**g**)). The effects of other OmpX-derived peptides on the SurA spectrum are shown in Supplementary Figs. 9, 10. **i** Model of SurA-WT in the core-P1 open state bound with peptides modelled in at the two binding hotspots. A short polypeptide sequence from a neighbouring molecule in the crystal structure (PDB: 1M5Y[48]) (cyan) is located in the core domain binding crevice. A peptide identified by phage-display which binds to the SurA P1 domain (WEYIPNV)[58] (pink) is modelled in from the P1-WEYIPNV peptide crystal structure (PDB: 2PV1[59]). Source data are provided as a Source Data file.

and/or thermodynamic signatures of binding to the core and P1 SurA binding hotspots, suggesting different (but as yet not fully understood) sequence specificities at each site.

## Binding of an OmpX-derived peptide promotes an activated open state

Binding of WEYIPNV to SurA-WT[55,56], and the gain-of-function mutant SurA-S220A[51], are known to promote SurA core-P1-open states (Fig. 1b). Residue S220 is located at the core-P1 interface[48] (Supplementary Fig. 14a), and results in the suppression of OM assembly defects caused by BamA/B mutations[55,56], implying that activation of SurA involves release of the P1 domain from the core domain[52]. Binding of the peptide WEYIPNV to SurA-WT causes substantial CSPs in the P1 domain, as well as the broadening of the M231 peak described above, as expected given the known binding specificity of WEYIPNV for this domain[59] (Fig. 4a, Supplementary Figs. 13, 15). In addition, CSPs are observed for peaks corresponding to residues across the whole core domain, except for residues in the core binding site mapped above using OmpX-derived peptides (Fig. 4a, Supplementary Fig. 13). Identical shifts in these core domain peaks are observed for SurA-S220A, indicating that these peaks report on the same conformational change in both cases (Fig. 4b, Supplementary Fig. 13g), and provide an NMR fingerprint for core-P1 opening. In addition to the changes observed for residues lining the core and P1 binding sites (Supplementary Fig. 9), binding of the OmpX-derived peptide QMN (OmpX β2), resulted in CSPs in identical (or similar) directions to those for the SurA-S220A variant (Fig. 4c, Supplementary Fig. 14). Combined, these results not only demonstrate that QMN peptide binds to SurA-WT in the identified cleft in the core domain, but that binding results in conformational changes associated with core-P1 opening, mirroring those observed for the S220A variant (Supplementary Fig. 14).

## Ar-X-Ar motifs are vital for specific binding to the SurA hotspots

To investigate the role of aromatic-containing motifs in SurA binding to the core and P1 binding sites, we produced an Ar-X-Ar-containing peptide from the sequence of a de novo designed transmembrane β-barrel (TMB), named MVK (TMB2.3 β3)[82]. Despite TMB2.3 and SurA having no evolutionary history, we observed interactions of MVK (TMB2.3 β3) with SurA-WT in both the core and P1 binding sites (Supplementary Fig. 16). These results provide additional supporting evidence that both the P1 and core binding sites recognise Ar-X-Ar motifs that are enriched in OMP sequences[57].

We next substituted aromatic residues within the Ar-X-Ar motif (YRY) of the OmpX-derived QMN (β2) peptide (QMN$_{YRY}$) with Ala, both separately (QMN$_{ARY}$, QMN$_{YRA}$), and together (QMN$_{ARA}$) (Supplementary Fig. 17, Supplementary Table 2). Peptides with a single Ala substitution removed the interaction with the core binding site and reduced peptide interaction at the P1 site (Supplementary Fig. 17a–f), while interactions with both sites were abolished by the double Ala substitution (Supplementary Fig. 17g, h). Finally, we recorded NMR spectra of SurA-WT in the presence of two truncated QMN (β2)

peptides: one comprising the N-terminal eight residues of QMN (QMNKMGGF - QMN$^{N-term}$), which lacks an Ar-X-Ar motif, and one comprising the C-terminal seven residues, which contains the Ar-X-Ar motif (NLKYRYE - QMN$^{C-term}$). As expected, QMN$^{N-term}$ that contains no Ar-X-Ar motif, had little effect on the spectrum of SurA-WT, consistent with the importance of Ar-X-Ar in mediating affinity (Fig. 5a, b). By contrast, QMN$^{C-term}$ demonstrated binding at the core site, but with binding largely abolished at the P1 site (Fig. 5c, d). Intriguingly, the CSPs shown to be indicative of the core-P1 conformational change when full-length QMN (β2) binds (Supplementary Fig. 14) are absent for QMN$^{C-term}$ binding (Fig. 5d), suggesting that whilst Ar-X-Ar motifs mediate binding, their flanking sequences are also important for relaying information to SurA when bound. Taken together, these results highlight the crucial role of Ar-X-Ar motifs in OMP recognition at both binding hotspots, and highlight the different sequence specificities at the two binding sites.

## Discussion

The OMP chaperone, SurA, plays a key role in cell envelope homoeostasis[33,34] and in promoting bacterial growth and virulence[47,83]. Yet, despite recent progress in mapping SurA domain dynamics (using smFRET[56], SANS[53], and NMR[55]) and client binding (using cross-linking MS[54,56], HDX-MS[56], and SANS[54]), the molecular details of how SurA binds its substrates and releases them to BAM for folding remain poorly understood. Here, we used methyl-TROSY NMR to map regions of the chaperone responsible for client recognition and binding in residue-specific detail. Previous models suggested that the affinity between the unfolded OMP client and SurA is mediated by diffuse binding to the chaperone surface involving the core/P1 (and potentially P2) domains[54,56]. In addition to the possibility of weak, diffuse binding, our NMR data, generated using different OMP sequences and OMP-derived peptides, revealed two OMP-binding hotspots, one located in the crevice between the two lobes of the N-domain of the core, and the other in the P1 domain. Differences (>100-fold) in affinity of short peptides derived from OmpX compared with the intact protein highlight the potential importance of both the binding hotpots and diffuse, weak interactions across the binding surface to enhance binding affinity. Further, from the perspective of the OMP client, we show that aromatic-containing motifs are important for OMP recognition at both sites, but that different patterns of aromatic residues and/or their flanking residues alter the specificity for each site. Ar-X-Ar motifs are enriched in OMPs[57], and are often found at the membrane-water interface in native OMP structures[6,7]. The specificity of SurA for these motifs helps explain how the chaperone can bind many different OMPs that exhibit a large diversity of sequence, size, and structures (Supplementary Fig. 7).

Recent AlphaFold-based predictions of the interaction between unfolded OmpA and SurA also suggested that clients bind in the core domain binding site we identified here[84]. These models suggested that a tyrosine (Y128) located at the base of the SurA core binding site forms π-π interactions with aromatic residues in OmpA. This prediction is consistent with the crucial importance of aromatic residues in the

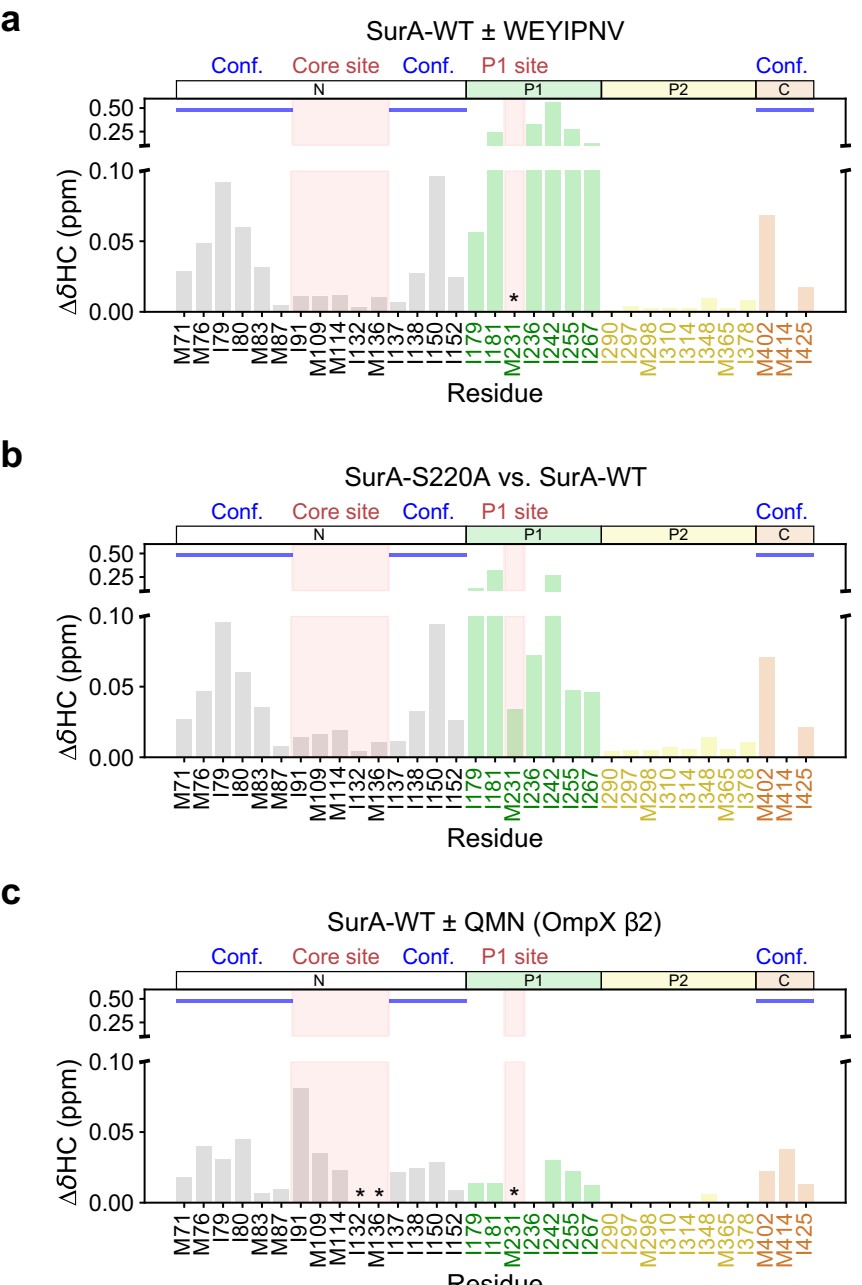

**Fig. 4 | Binding of an OmpX-derived peptide promotes core-P1 open 'activated' SurA.** Chemical shift differences between (**a**) SurA-WT + WEYIPNV and SurA-WT alone, (**b**) SurA-S220A and SurA-WT, and (**c**) SurA-WT + QMN peptide (OmpX β2) and SurA-WT alone. Peaks which were broadened beyond detection in peptide-containing samples are indicated with an asterisk. Conf.: residues indicative of conformational change (Supplementary Fig. 14). Samples contained 5 μM SurA or SurA-S220A, in 5 mM EDTA, 20 mM Tris-HCl, pH 8, at 25 °C. In samples containing the WEYIPNV or QMN peptides, these were present at 50 μM or 200 μM, respectively. Source data are provided as a Source Data file.

substrate for binding at this site, as we show via substitution of Tyr with Ala in the Ar-X-Ar motif of the OmpX-derived QMN peptide (Supplementary Fig. 17). The two binding hotspots may have different roles in chaperone function, for instance, promoting core-P1 opening, client expansion, client release and/or delivery to BAM. Additionally, by enabling different regions of the same unfolded OMP chain to anchor to each site, the two binding hotspots may facilitate vectorial delivery of unfolded OMPs to BAM for folding. Exactly how OMPs are released from SurA and delivered to BAM remain open questions, but changes in conformational dynamics of SurA on BAM binding[50], together with the weak affinities of SurA-OMP interactions[49,81,85], and ultimately the

driving force of the onset of OMP folding into the OM[86] are all likely to play a role.

Combining the evidence presented, we propose a refined model for SurA:OMP chaperone activity (Fig. 6). In the absence of its OMP clients, SurA is found predominantly in its auto-inhibited core-P1 closed state[52,53]. Binding of unfolded OMPs via their aromatic motifs to the chaperone promotes opening of the core-P1 interface to create a more active state. Given that the core binding site is accessible in core-P1 closed models[48,53], the results suggest that auto-inhibition of core activity by P1 occurs via suppression of conformational dynamics of the two lobes that comprise the core domain binding site, rather than

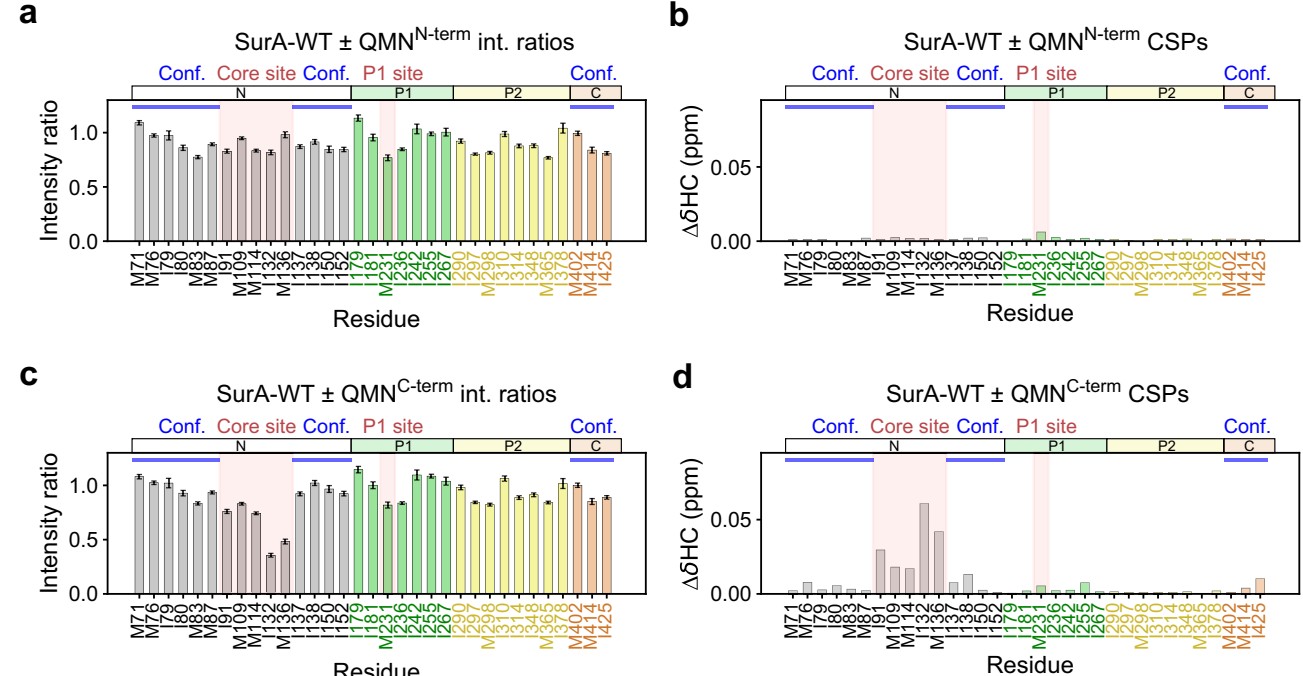

**Fig. 5 | Effects of truncation of the OmpX-derived QMN (b2) peptide on the methyl-TROSY NMR spectrum of SurA-WT.** Intensity ratio (**a**, **c**) and CSP (**b**, **d**) plots are shown for SurA-WT bound to (**a**, **b**) the N-terminal 8 residues of QMN (QMN$^{N-term}$ - QMNKMGGF), and (**c**, **d**) the C-terminal 7 residues of QMN (QMN$^{C-term}$ - NLKYRYE). Peaks which were broadened beyond detection in peptide-containing samples are indicated with an asterisk. Conf.: residues indicative of conformational change (Supplementary Fig. 14). Samples contained 5 µM SurA ± 200 µM peptide, 5 mM EDTA, 20 mM Tris-HCl, pH 8, at 25 °C. int: intensity. Signal to noise ratios were used for the calculation of errors in peak intensities in (**a**, **c**) (Methods). Source data are provided as a Source Data file.

direct blocking of this site from OMP binding. Consistent with this, crosslinking P1 to the core domain leads to OMP assembly defects in vivo[52] and reduced crosslinking to OMP substrates in vitro[54]. Relief of the auto-inhibition of the core domain by P1 (in which the P2 domain may play a role[49,52]) could involve increased flexibility and dynamics within the core domain's binding crevice, and/or clamp-like motions between the two lobes of the N-domain. Increased flexibility induced by mutations is known to enhance the activity of other ATP-independent chaperones, such as Spy[87], and clamp-like binding sites are commonly found in both ATP-dependent (e.g., Hsp70, and Hsp90 families)[88], and ATP-independent chaperones (e.g., Skp, prefoldin)[88,89]. Activation of ATP-independent chaperones often involves order-to-disorder transitions and/or changes in oligomeric state[90,91], while for ATP-dependent chaperones, activation depends upon structural rearrangements caused by the binding, hydrolysis and/or release of nucleotides[92,93]. By contrast, our data reveal that in the case of SurA, it is interaction with substrates which induces domain rearrangements that activate the chaperone.

Inhibition of SurA activity is a promising strategy for combating Gram-negative pathogen growth, virulence, and resistance to current antibiotics. However, given previous models based on diffuse binding to the chaperone surface, how this could be achieved was unclear. Our identification of two binding hotspots for OMP clients in the core and P1 domains reveal these sites as excellent potential targets for small molecules, which could potentially interfere simultaneously with chaperone binding, activation, substrate remodelling, and OMP delivery to BAM.

## Methods
### Bioinformatics analysis of SurA domain conservation
To assess the level of conservation of PPIase domains in SurA domain architectures, we obtained 14,244 SurA homologues from the InterPro database (InterPro family IPRO15391). We identified core only

homologues in several steps. First, homologues were filtered by selecting those which had no predicted PPIase domain InterPro families (IPR046357, IPR000297, IPR023058, or IPR023034) annotated in their UniProt entries (1159 sequences). Next, we used Alpha-Fold structural predictions for these sequences to separate homologues which contain unannotated PPIase domains from genuine core only homologues. Of the 1159 sequences, 1120 predicted structures were found in the EBI-hosted AlphaFold database (https://alphafold.ebi.ac.uk/) (AF_<uniport_id>_F1_model_v3.pdb files). We generated structural predictions for the remaining 39 homologues using Alphafold-Multimer (v2.1.0) installed on a local workstation and the reduced databases as described at https://github.com/deepmind/alphafold. We then used DSSP predictions of the secondary structure for filtering as, unlike the core domain, PPIase domains all have a topology that includes three consecutive β-strands. Successful filtering was manually confirmed by visualisation in PyMOL. By evaluating these AlphaFold2-generated structures we found that only 73/1159 of these sequences were predicted to be true core domain only homologues (Source Data file). The majority of these are from the γ-protobacterial class in the genus *Halomonas* (49/73). Therefore, only ~0.5% of the 14,244 SurA homologues in the InterPro database are predicted to be core domain only homologues (Fig. 1c). Analysis was carried out using in-house Python 3 scripts and made use of the Biopython (v1.79)[94,95] library.

### Conservation analysis of SurA homologues
To obtain a set of SurA homologues with a three domain architecture, sequences with lengths between 400 and 450 residues were extracted from the InterPro SurA_N family (IPRO15391)[96] (8250/14,225 homologues). These sequences were then clustered with CD-HIT[97] using a 40% sequence identity threshold to give 303 representative sequences. A multiple sequence alignment (MSA) of these sequences was obtained with Clustal Omega[98] with default settings accessed via

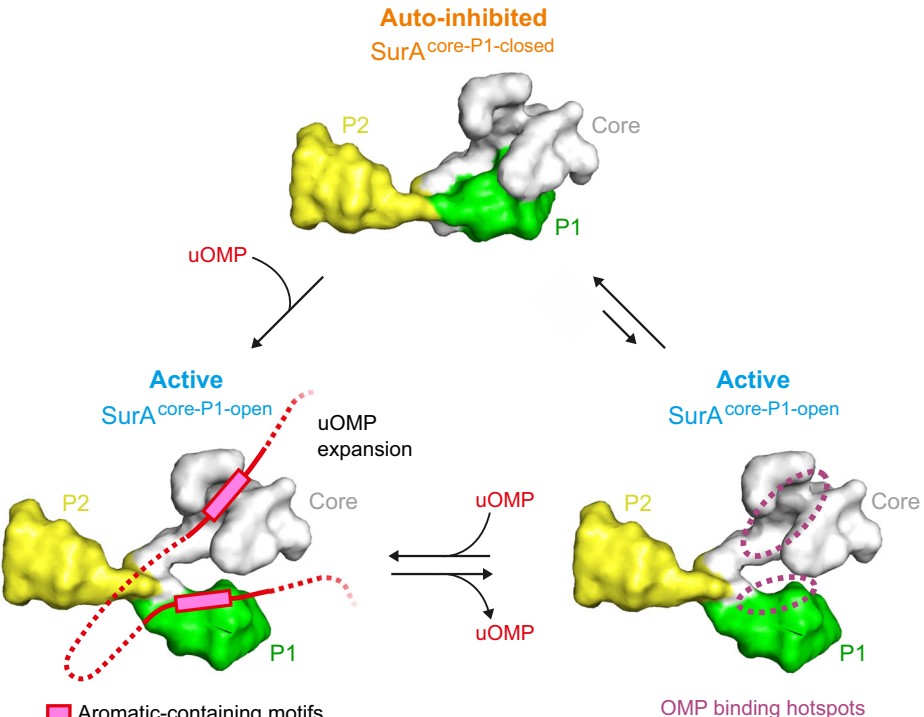

**Fig. 6 | Proposed activation mechanism for OMP binding to SurA.** In the absence of substrate SurA is predominantly in an auto-inhibited core-P1 closed conformation (top) with a minor population (-25%[53]) of an active core-P1 open conformation (right). Binding of OMP substrates (red) to the two identified binding hotspots in the core and P1 domains enables activation of the chaperone and leads to OMP expansion (left). These dual binding sites recognise aromatic containing motifs (e.g., Ar-X-Ar) that are enriched in OMP sequences[57]. Whether it is OMP binding to the core site, P1 site, or both, that leads to chaperone activation is currently not clear, though our results implicate P1 binding in this process (Fig. 4, Supplementary Fig. 13). Binding sites of both hotspots are accessible in the core-P1 closed state suggesting that auto-inhibition is due to repression of conformational dynamics of the core domain, rather than steric blocking, and that sequences linking binding motifs in intact unfolded OMP clients are important for chaperone activation. Expansion of the substrate and prevention of locally collapsed regions in the unfolded chain by activated SurA suggests a mechanism for delivery of the 'linearised' client to BAM for its vectorial folding into the OM[54,63].

---

JalView v2.11.2.5[99]. This MSA was used to generate per residue conservation scores via the ConSurf webserver[100,101]. Residues with a ConSurf score of 7/9 or higher were considered highly conserved.

## Plasmid constructs

The expression plasmid pET28b containing the mature SurA sequence (residues 21–428) preceded by an N-terminal 6x His-tag and thrombin-cleavage site (pSK257)[102] was a kind gift from Daniel Kahne (Harvard University, USA). The thrombin-cleavage site was changed to a TEV-cleavage site using mutagenesis, yielding SurA with the N-terminal sequence MGSS(H)$_6$SSGENLYFQG. SurA domain deletion variants SurA-ΔP1, SurA-ΔP2, and SurA-core were constructed by deleting codons for residues 172–280, 281–389 or 172–389, respectively, as described previously[49]. The expression plasmid for the mature sequence of tOmpA (residues 22–192) in pET11a[103] was kindly provided by Karen Fleming (John Hopkins University, USA). Codon-optimised synthetic genes (Eurofins) of the mature sequences of OmpX (residues 24–171) and OmpF (residues 23–362) were cloned into pET11a (Novagen) between the NdeI (5′) and BamHI (3′) restriction sites. pET29b+ expression plasmids encoding the Cys-OmpX-Cys, Cys-OmpX, and OmpX-Cys constructs used in smFRET experiments were obtained from TWIST bioscience. The Cys-OmpX-Cys construct contained the N-terminal sequence Met-Gly-Ser-Cys followed by the codon-optimised mature OmpX sequence (residues 24–171) and an additional C-terminal Cys residue[63]. The Cys-OmpX[50] and OmpX-Cys single Cys variants of the Cys-OmpX-Cys construct lacked the C-terminal and N-terminal Cys residues, respectively. The Cys-tBamA construct for use in MST experiments, was also obtained from TWIST bioscience in a pET29b+ expression vector. This construct contained the N-terminal sequence Met-Gly-Ser-Cys followed by the codon-optimised mature tBamA sequence (residues 425–810). The two natural Cys residues in tBamA were mutated to Ser (C690S/C700S). To create tBamA-pET11a, residues 425–810 of BamA were amplified by PCR, using plasmid BamAB-pETDUET-1 (kindly donated by Susan Buchanan (NIH, USA)) as the template, and the resultant product was then ligated into pET11a between NdeI and BamHI restriction sites. The plasmids containing SurA isoleucine and methionine point mutants (Supplementary Table 3) used in NMR peak assignment experiments were obtained from TWIST bioscience cloned into pET29b+. All mutagenesis was performed using Q5 site-directed mutagenesis (NEB). The protein sequences of expressed OMPs and SurA variants are provided in the Source Data file.

## Expression and purification of unlabelled SurA-WT and SurA variants

Plasmids encoding wild-type SurA or SurA variants were transformed into BL21(DE3) cells (Stratagene). Cells were grown in LB medium supplemented with 40 μg/mL kanamycin at 37 °C with shaking (200 rpm) until an OD$_{600}$ of -0.6 was reached. The temperature was subsequently lowered to 20 °C, and expression induced with 0.4 mM IPTG. After -18 h, cells were harvested by centrifugation, resuspended in 20 mM Tris-HCl, pH 7.2, 150 mM NaCl, 20 mM imidazole, containing EDTA-free protease inhibitor tablets (Roche), and lysed using a cell disrupter (Constant Cell Disruption Systems). The cell debris was removed by centrifugation (20 min, 4 °C, 39,000 × g), and the lysate was diluted to 0.5–1 L volume with 20 mM Tris-HCl, pH 7.2, 150 mM NaCl, and applied to 2 × 5 mL HisTrap columns (Cytiva). The columns were washed with 20 mM Tris-HCl, pH 7.2, 150 mM NaCl and 20 mM

imidazole and SurA was eluted with 20 mM Tris-HCl, 150 mM NaCl, 500 mM imidazole, pH 7.2. The eluate was dialysed extensively against 20 mM Tris-HCl, pH 8.0, before being concentrated to ~200 μM with Vivaspin 20 concentrators (Sartorius; 5 kDa MWCO), aliquoted, snap-frozen in liquid nitrogen and stored at −80 °C. Protein concentrations were determined spectrophotometrically (Supplementary Table 4).

## Labelling of Cys-OmpX-Cys variant

Purified Cys-OmpX-Cys was covalently labelled stochastically with Alexa Fluor 488 and ATTO 565 dyes via maleimide chemistry. A sample containing 423 μM Cys-OmpX-Cys in 25 mM Tris-HCl, 6 M Gdn-HCl, pH 7.2, was incubated with 10 mM DTT for 30 min. This sample was subsequently buffer exchanged into 25 mM Tris-HCl, 6 M Gdn-HCl, pH 7.2 (that had been sparged for 15 min with nitrogen gas) using Zeba spin desalting columns (Thermo Fisher Scientific). Alexa Fluor 488 C5 maleimide (Thermo Fisher Scientific) (10 mM dissolved in DMSO) and ATTO 565 maleimide (ATTO-TEC GmbH) (10 mM dissolved in DMSO) were immediately added to the OmpX sample to final concentrations of 635 μM and 1.17 mM, respectively. The total sample volume was 500 μL. The labelling reaction was incubated at 25 °C for 30 min only to avoid non-specific labelling. The reaction was quenched with 10 mM DTT and the sample then loaded onto a Superdex Peptide 10/300 column (GE Healthcare) equilibrated with 3 M Gdn-HCl, 25 mM Tris-HCl, pH 7.2 to remove the excess free dye. Samples were collected every 1 mL and peak protein fractions tested for dye labelling using a Nanodrop 2000 (Thermo Fisher Scientific). Samples containing double-labelled Cys-OmpX-Cys were snap-frozen using liquid nitrogen and stored at −80 °C until required.

## Labelling of Cys-OmpX and OmpX-Cys variants

Cys-OmpX was labelled with Alexa Fluor 488 as previously described[50]. For labelling of Cys-OmpX with ATTO 565 and OmpX-Cys with both Alexa Fluor 488 and ATTO 565, samples containing 300 μM protein were first incubated with 10 mM DTT for 30 min. Samples were then buffer exchanged into 25 mM Tris-HCl, 6 M Gdn-HCl, pH 7.2 (that had been sparged for 15 min with nitrogen gas) using Zeba spin desalting columns (Thermo Fisher Scientific). Either Alexa Fluor 488 C5 maleimide (Thermo Fisher Scientific) (10 mM dissolved in DMSO) or ATTO 565 maleimide (ATTO-TEC GmbH) (10 mM dissolved in DMSO) as appropriate were immediately added to final concentrations of 450 μM (1.5x excess) and incubated at 25 °C for 30 min. The reaction was quenched with 10 mM DTT, and the sample then loaded onto a Superdex Peptide 10/300 column (GE Healthcare) equilibrated with 3 M Gdn-HCl, 25 mM Tris-HCl, pH 7.2 to remove the excess free dye. Samples were collected every 1 mL and peak protein fractions tested for dye labelling using a Nanodrop 2000 (Thermo Fisher Scientific).

## smFRET data collection

Samples were measured at a volume of 100 μL in an 18 well sample chamber (81817, Ibidi) with the focal spot 50 μm above the cover glass on a custom-built confocal microscope (Supplementary Methods) at room temperature (~21 °C). Laser powers measured before the objective lens of the two lasers used for donor and acceptor excitation were an average of 75 μW and 35 μW, respectively. smFRET of samples was measured at a concentration of ~75 pM OmpX in measurement buffer (20 mM Tris-HCl, 0.1 M urea, 0.02% (v/v) Tween-20, pH 8.0) with 1 mM aged Trolox ((±)-6-hydroxy-2,5,7,8-tetramethylchromane-2-carboxylic acid) to prevent photobleaching of the sample[104]. Aged Trolox was prepared by adding 1 mM of Trolox to measurement buffer and incubating the sample overnight at room temperature on a rotary shaker to dissolve before being filtered using a 0.22 μm pore size membrane. Tween-20 (0.02% v/v) was added to prevent non-specific absorption to the glass slide. In experiments monitoring FRET-labelled OmpX in the presence of SurA variants, a concentration of 30 μM SurA-WT or SurA-core was used. Each sample was freshly prepared and measured in 1 h with

acquisitions with a total of 5 measurements for apo-OmpX (0.1 M urea) and 6 measurements for OmpX + SurA-WT and OmpX + SurA-core. apo-OmpX in 4 M urea was measured for 1 h.

## smFRET data analysis

The ptu files from the SymPhoTime64 software were converted into HDF5 data files using phconvert v0.9 (available at https://github.com/Photon-HDF5/phconvert). Each of these files was then merged to create one file for each condition. These merged files were then analysed using the FRETBursts v0.7.1 python package[105]. Firstly, the background was independently calculated for every 60 s period of measurement. Following the background calculation, a burst search was performed, selecting bursts with a minimum threshold of 3x the background signal in the donor and acceptor channels, a minimum total burst size of 50 photons in the donor and acceptor channels from the donor excitation. Bursts were then corrected[106] through the FRETBursts software[105] using the following values: donor leakage = 0.14, direct excitation = 0.14, gamma factor = 1.0 and beta factor = 1.4. Bursts were then filtered by removing events with an S > 0.7 and <0.3 and an ALEX-2CDE[107] filter with a cut off of 95 (as implemented in FRETBursts) to remove bursts with photobleaching, excessive photophysics or potential multi-molecule events.

BVA[68] was used to assay for dynamics on the ms timescale occurring during bursts. This analysis compares the expected variance in $E_{raw}$ based on shot noise against the measured variance in $E_{raw}$ in each individual burst ($s_i$). BVA was performed in the FRETBursts environment. To calculate $s_i$, we segmented each burst into consecutive windows of 5 photons, and calculated the standard deviation of all windows within the burst. To help visualise the average $s_i$ at each $E_{raw}$ we segmented the $E_{raw}$ axis into bins with a width of 0.05.

The apparent fluorescence lifetime of the donor fluorophore in the presence of the acceptor ($\tau_{D(A)}$) within each FRET burst was calculated by taking the mean microtime (corresponding to donor excitation and donor emission) and subtracting the mean microtime of the instrument response function[108]. The lifetime of the donor molecule in the absence of the acceptor ($\tau_{D(0)}$) was calculated in the same fashion, but for OmpX molecules which were only labelled with a donor fluorophore (i.e., a S value >0.9). The static and dynamic FRET lines were calculated using the FRETlines python package version e099b1a (available at https://github.com/Fluorescence-Tools/FRETlines)[109].

## Expression and purification of SurA variants for NMR spectroscopy

Plasmids containing wild-type SurA or SurA variant genes were transformed into BL21(DE3) cells (Stratagene). A daytime starter culture of 5 mL LB, supplemented with 40 μg/mL kanamycin, was grown from a single colony for ~6 h at 30 °C to an $OD_{600}$ of ~0.5. This culture was centrifuged, resuspended in 100% $D_2O$ and added to NMR media (Supplementary Table 5) supplemented with 40 μg/mL kanamycin, in 250 mL baffled conical flasks, for overnight growth at 30 °C. For a typical 100 mL culture the volume of the overnight culture was 24 mL. The following morning the remaining 76 mL of NMR media supplemented with 40 μg/mL kanamycin was added to the culture and grown at 37 °C until an $OD_{600}$ of ~0.5–6 was reached (~4–6 h). Reagents for Ile and Met labelling α-ketobutyric acid and L-methionine (Supplementary Table 6) at 70 mg/L and 250 mg/L were added and the culture grown for 1 h before expression was induced with the addition of 0.4 mM IPTG. After 6 h of expression the cells were harvested by centrifugation. We found that expression with 2x M9 minimal media (Supplementary Table 5) to increase buffering capacity led to an approximately 1.5–2-fold increase in yields[110]. Purification was performed as for unlabelled SurA with some variation in the protocol for the purification of Ile and Met variants for assignment. For the latter, all buffers were the same, but cells from these small-scale expression cultures (50 mL) were lysed by sonication, resuspended in 1 mL and

His$_6$ SpinTrap columns (Cytivia) were used for the purification according to manufacturer's instructions.

## Expression and purification of OMPs

OMPs (OmpX, tOmpA, OmpF, tBamA, Cys-tBamA, Cys-OmpX-Cys, Cys-OmpX and OmpX-Cys) were purified using a method adapted from McMorran et al.[111]. Briefly, *E. coli* BL21[DE3] cells (Stratagene) were transformed with a plasmid containing the gene sequence of the mature OMP. Overnight cultures were subcultured and grown in LB medium (500 mL) supplemented with carbenicillin (100 µg/mL), at 37 °C with shaking (200 rpm). Protein expression was induced with IPTG (1 mM) once an OD$_{600}$ of 0.6 was reached. After 4 h the cells were harvested by centrifugation (5000 × *g*, 15 min, 4 °C). The cell pellet was resuspended in 50 mM Tris-HCl pH 8.0, 5 mM EDTA, 1 mM phenylmethylsulfonyl fluoride, 2 mM benzamidine, and the cells were subsequently lysed by sonication. The lysate was centrifuged (25,000 × *g*, 30 min, 4 °C) and the insoluble material was resuspended in 50 mM Tris-HCl pH 8.0, 2% (*v/v*) Triton X-100, before being incubated for 1 h at room temperature, with gentle agitation. The insoluble material was pelleted (25,000 × *g*, 30 min, 4 °C) and the inclusion bodies washed twice by resuspending in 50 mM Tris-HCl pH 8.0 followed by incubation for 1 h at room temperature with gentle agitation, and then collected by centrifugation (25,000 × *g*, 30 min, 4 °C). For the OmpX, tOmpA, OmpF, and tBamA constructs, the inclusion bodies were solubilised in 25 mM Tris-HCl, 6 M Gdn-HCl, pH 8.0 and centrifuged (20,000 × *g*, 20 min, 4 °C). The supernatant was filtered (0.2 µM syringe filter, Sartorius) and the protein was purified using a Superdex 75 HiLoad 26/60 gel filtration column (GE Healthcare) equilibrated with 25 mM Tris-HCl, 6 M Gdn-HCl, pH 8.0. Peak fractions were concentrated to 500 µM using Vivaspin 20 (5 kDa MWCO) concentrators (Sartorius), and the protein solution was snap-frozen in liquid nitrogen and stored at −80 °C. Protein concentrations were determined spectrophotometrically (Supplementary Table 4).

## NMR spectroscopy

All NMR experiments were performed on a 950-MHz Bruker Ascend Avance III HD NMR spectrometer equipped with either a Bruker 3 mm TCI or 5 mm TXO triple-resonance cryogenically cooled probe. Peptide binding experiments were performed in 5 mM EDTA, 20 mM Tris-HCl, pH 8.0 with 5 % (*v/v*) D$_2$O at 25 °C. Binding experiments using full-length OMP sequences also contained 0.8 M urea. Data was acquired for ~16 h with the exception of OMP-containing experiments which were acquired for ~5 h. Methyl-TROSY NMR experiments used a band-selective optimised-flip-angle short-transient experiment[112] (2D $^1$H-$^{13}$C SOFAST-HMQC)[70]. Pulse sequences were obtained using the NMRlib[113] plugin for TopSpin. Data was processed with NMRPipe[114] and analysed with CcpNmr Analysis v2.5[115]. Further analysis and all plotting was carried out using in-house Python 3 scripts using the Matplotlib (v3.2.2)[116], NumPy (v1.18.5)[117], Nmrglue (v0.8)[118], and Biopython (v1.79)[94,95] libraries. Signal to noise ratios were used for the calculation of errors in peak intensities, with the noise floor obtained for each spectrum using NMRPipe (specStat.com)[114]. Errors in the intensity ratios for each peak between two spectra were obtained by propagating the errors according to:

$$\delta R = |R| * \sqrt{\frac{\delta X}{X}} + \sqrt{\frac{\delta Y}{Y}} \tag{1}$$

where δR is the error in the intensity ratio, |R| is the value of the intensity ratio, X and Y are the intensity values for the peak in the first and second spectrum, respectively, and δX and δY are the noise floor values for the first and second spectrum, respectively.

For the calculation of CSPs, the shifts in the proton and carbon dimensions were scaled according to the relative gyromagnetic ratios of the nuclei:

$$CSP = \sqrt{\Delta H^2 + (0.251 \times \Delta C)^2} \tag{2}$$

where ΔH and ΔC are the changes in proton and carbon chemical shift, respectively. Z-scores for intensity ratio and CSP data were calculated using:

$$Z_i = \frac{x_i - \mu}{\sigma} \tag{3}$$

where $x_i$ is the value for a particular peak, and µ and σ are the mean and standard deviation for all peaks, respectively.

## Assignment of SurA Ile-Cδ1 and Met-Cε resonances in NMR spectra

For assignment of SurA-WT Ile and Met residues (43 in total) each residue was individually mutated (Supplementary Table 3), and the effect of the mutation assessed by methyl-TROSY NMR. Samples contained 5–70 µM SurA variant. Assignment experiments (2D $^1$H-$^{13}$C SOFAST-HMQCs) for Ile or Met variants were performed in 20 mM sodium phosphate, 50 mM NaCl, pH 7.5 (at 35 °C), or 20 mM Tris-HCl, pH 8 (at 25 °C), respectively. Two pairs of peaks (I70/I280, I100/I103) were overlapped. One pair of peaks in the P1 domain (I202/I259) could not be unambiguously assigned. One pair of severely broadened peaks in the core domain (M46/M400) could also not be unambiguously assigned. Further one peak in the P1 domain (I217) was also severely broadened and not included in analysis. Finally, the peak for I75 could not be separated from that for I202/I259 in many experiments, so it was also not included in analyses. Peaks for I70 and M400 appear in new locations in experiments in which core-P1 open states are populated (SurA-WT + WEYIPNV and SurA-S220A) or the P1 domain is deleted (SurA-core and SurA-ΔP1). These were assigned by performing experiments in which WEYIPNV (50 µM) was added to the I70V and M400A SurA variants. The $^1$H chemical shift assignments were referenced using 2,2-dimethyl-2-silapentane-5-sulphonic acid (DSS) at 25 °C as a standard. The $^{13}$C chemical shifts were referenced indirectly to DSS. NMR assignments have been deposited in the Biological Magnetic Resonance Bank (BMRB) with Entry ID 52180.

## Peptide synthesis

Peptides from internal regions of OMPs were modified with N-terminal acetylation and C-terminal amidation and were either ordered from GenScript (USA) or produced in-house. All amino acids, Rink amide resin, N,N'-diisopropylcarbodiimide (DIC), 2-cyano-2-(hydroxyimino)acetate (Oxyma), trifluoroacetic acid, triisopropylsilane and 2,2'-(Ethylenedioxy)diethanethiol (DODT) were purchased from Fluorochem or Merck. All amino acids were *N*-Fmoc protected and side chains were protected with either *t*Bu (Asp, Glu, Ser, Thr, Tyr), Trt (Asn, Gln, Cys, His), Boc (Lys, Trp) or Pbf (Arg). Piperidine and acetic anhydride were purchased from Fisher. ACS grade N,N-dimethylformamide (DMF) and HPLC grade Acetonitrile from Merck were used during the synthesis and analysis. The identity of the peptide was confirmed by high-resolution mass spectrometry on Bruker Maxis Impact spectrometer using electrospray ionisation. The purity was determined on Agilent 1290 Infinity II system using Ascentis peptide column and gradient 5–95% of acetonitrile in water with 0.1% (v/v) trifluoroacetic acid additive at 0.5 ml/min for 10 min. The peptide synthesis was performed in microwave assisted Liberty Blue CEM peptide synthesiser equipped with HT loader at 0.1 mmol scale on Rink-amide resin (loading 0.48 mmol/g). Standard pre-programed coupling and deprotection cycles were applied. The deprotection was achieved using 20% (v/v) piperidine in DMF with microwave heating at 90 °C for 100 s, followed by three DMF washing steps. The couplings were performed using 5 eq. of Fmoc-protected

amino acid, 5 eq. DI and 5 eq. Oxyma in DMF with microwave heating at 90 °C for 3 min followed by two DMF washing steps. The N-terminal acetylation was performed using 10% (v/v) acetic anhydride in DMF with microwave heating at 60 °C for 3 min followed by two DMF washing steps. The resin with synthesised peptide was then transferred to SPS tube, washed three times with 10 Ll dichloromethane, two times with 10 mL of diethylether and dried under vacuum for 30 min. To deprotect the side chains and cleave the peptide from the resin, the resin was incubated on a rotator for 3 h with 10 ml of cleavage mix (92.5% (v/v) trifluoroacetic acid, 2.5% (v/v) water, 2.5% ($v/v$) triisopropylsilane, 2.5% ($v/v$) DODT) and filtered. The filtrate was concentrated to $ca.$ 1 mL under a stream of nitrogen and the peptide was precipitated by addition of 10 mL of ice-cold diethylether and isolated by centrifugation ($4226 \times g$ for 5 min). The precipitate was resuspended in 10 mL of ice-cold diethylether and isolated by repeating the centrifugation step. After decanting the diethylether, the precipitate was allowed to dry for 30 min, dissolved in 5Ll of 1% ($v/v$) acetic acid and freeze dried.

## Peptide purification

Peptides were dissolved in 4–10 mL of 1:1 mixture of acetonitrile and water and purified using Agilent 1260 infinity system equipped with UV detector and fraction collector on Kinetex EVO 5 µm C18 100 Å 21.2 × 250 mm reverse phase column. 1–4 mL of the peptide solution was injected and 25 min gradient of 10–40% (v/v) acetonitrile in water with 0.1% (v/v) formic acid additive was run at 10 ml/min. The fractions containing peptide were pooled and freeze dried.

## Isothermal calorimetry (ITC)

ITC experiments were performed on a MicroCal iTC200 instrument in duplicate at 25 °C. The buffer used in all experiments was 20 mM Tris-HCl, pH 8. For binding experiments between SurA-WT and the peptide WEYIPNV, concentrations of 20 µM SurA-WT and 200 µM in the sample cell and syringe were used, respectively. For binding experiments between SurA-WT and OmpX-derived peptides QMN and KHD, concentrations of 100 µM SurA-WT and 1000 µM QMN/KHD in the sample cell and syringe were used, respectively. Other Ar-X-Ar containing OmpX-derived peptides were not soluble at these concentrations. 20 injections ($1 \times 0.5$ µL and $19 \times 2$ µL) of peptides were added successively to the sample cell with a spacing of 150 s and a stirring rate of 750 rpm. Reference experiments in which SurA-WT was absent from the sample cell were performed and subtracted from the corresponding peptide binding data. The first data point was discarded in all experiments. Data were processed, analysed and fitted using OriginLab ITC v1.25.5.

## Labelling of Cys-tBamA with Alexa Fluor 488

Purified Cys-tBamA was covalently labelled with Alexa Fluor 488 dye via maleimide chemistry. A sample containing 153 µM Cys-tBamA in 25 mM Tris-HCl, 6 M Gdn-HCl, pH 7.2, was incubated with 10 mM DTT for 30 min. This sample was subsequently buffer exchanged into 25 mM Tris-HCl, 6 M Gdn-HCl, pH 7.2 (that had been sparged for 15 min with nitrogen gas) using Zeba spin desalting columns (Thermo Fisher Scientific). Alexa Fluor 488 C5 maleimide (Thermo Fisher Scientific) (10 mg/mL dissolved in DMSO) was immediately added to the Cys-tBamA sample at a final concentration of 1.5 mM. The total sample volume was 260 µL. The labelling reaction was kept at 25 °C for 1 h then left overnight at 4 °C. The reaction was then loaded onto a Superdex Peptide 10/300 column (GE Healthcare) equilibrated with 6 M Gdn-HCl, 25 mM Tris-HCl, pH 7.2 to remove the excess free dye. Samples were collected every 1 mL and peak protein fractions tested for dye labelling using a Nanodrop 2000 (Thermo Fisher Scientific). Samples containing labelled tBamA were snap-frozen using liquid nitrogen and stored at −80 °C until required.

## Microscale thermophoresis (MST)

Alexa Fluor 488-labelled tBamA or OmpX was buffer exchanged into 8 M urea and 20 mM Tris−HCl, pH 8.0. A stock of 200 µM SurA-WT or SurA-core in 20 mM Tris−HCl, pH 8.0 was used to create a serial dilution for tBamA (40 µM–1.2 nM) and for OmpX (12.5 µM–3 nM). A 1 µM stock of Alexa Fluor 488-labelled tBamA or OmpX was first diluted to 200 nM with 20 mM Tris−HCl, pH 8.0, then added to SurA variant samples to give final concentrations of 100 nM tBamA/OmpX and 0.8 M urea in 20 mM Tris−HCl, pH 8.0. The samples were loaded into premium-coated capillaries (NanoTemper Technologies GmbH, Germany) and measured using a Monolith NT.115 (NanoTemper Tech.) at 25 °C. Data were fitted to a Hill equation:

$$S_{obs} = S_U + \left( \frac{(S_B - S_U)}{1 + \left(\frac{K_{d,app}}{[L]}\right)^n} \right) \tag{4}$$

where $S_{obs}$ is the observed signal; $S_U$ and $S_B$ are the signal of the unbound and bound state, respectively; $K_{d,app}$ is the apparent $K_d$; $L$ is the ligand, which in these experiments is SurA-WT or SurA-core, and $n$ is the Hill coefficient. Fits and plots were made with in-house Python 3 scripts and made use of the SciPy (v1.5.0) and Matplotlib (v3.2.2)[116] libraries.

## Reporting summary

Further information on research design is available in the Nature Portfolio Reporting Summary linked to this article.

## Data availability

Assignments for Ile and Met peaks in the 2D $^1$H-$^{13}$C methyl-TROSY NMR spectrum of SurA-WT have been deposited in the Biological Magnetic Resonance Bank (BMRB Entry ID: 52180). Raw smFRET data are available at the University of Leeds data repository (https://doi.org/10.5518/1423). AlphaFold 2 generated models of SurA homologues are available at https://doi.org/10.5281/zenodo.13150730. The source data underlying Figs. 1–5, and Supplementary Figs. 1–3, 5–11, and 13–17 are provided as a Source Data file. All other data are available from the corresponding author on request. Reference structures used in the work are 2PV1, 2PV2, 2PV3, 1M5Y, and 1QJ8. Source data are provided with this paper.

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

## Acknowledgements

We thank members of the Zhuravleva, Radford and Brockwell laboratories for helpful discussions. We also thank Arnout Kalverda for excellent technical support and Roman Tuma for helping with developing smFRET and its analysis. We acknowledge Wellcome for funding the Chirascan CD spectrometer was (094232/Z/10/Z), the Monolith NT.115 MST (105615/Z/14/Z) and ITC instruments (094232/Z/10/Z). We acknowledge Astbury Biostructrure Laboratory for access to the 950-MHz spectrometer which was funded by the University of Leeds. B.S. acknowledges support from the BBSRC (BB/T000635/1). A.N.C. acknowledges the support of a Sir Henry Dale Fellowship jointly funded by Wellcome and the Royal Society (Grant Number 220628/Z/20/Z) and a University Academic Fellowship from the University of Leeds. S.E.R. holds a Royal Society Professorial Fellowship (RSRP/R1/211057). We acknowledge funding from BBSRC for J.A.C. (BB/T008059/1) and M.W. (BB/ V003577/1). J.M.M. was funded by Wellcome (222373/Z/21/Z). We thank EPSRC (EP/N013573/1) and BBSRC (BB/V003577/1) for funding peptide synthesis. T.F. is supported by the European Regional Development Fund-Project (CZ.02.1.01/0.0/0.0/15_003/0000441) and the Czech Science Foundation (20-11563Y). For the purpose of Open Access, the authors have applied a CC BY public copyright licence to any Author Accepted Manuscript version arising from this submission.

## Author contributions

B.S. and J.A.C. designed, performed, and analysed the data for NMR and smFRET experiments, respectively. M.W. designed and synthesised peptides. J.M.M. generated SurA models. G.N.K. performed control experiments and provided laboratory support. I.W.M. performed and analysed ITC experiments. T.F. assisted with software for smFRET data analysis. A.J.W., D.J.B., A.N.C., S.E.R. and A.Z. supervised the research. The paper was written by B.S., J.A.C., S.E.R. and A.Z. with input and comments from all authors.

## Competing interests

The authors declare no competing interests.
