## [Peer Review File · Nature Communications]

Dual client binding sites in the ATP-independent chaperone SurAREVIEWER COMMENTS

Reviewer #1 (Remarks to the Author):

Schiffirin, Crossley and colleagues present in this manuscript a study of the periplasmic chaperone SurA. In particular they report an extensive study of potential binding sites of unfolded outer membrane proteins and peptides to SurA using NMR spectroscopy. They reveal various important residues in SurA in P1 and the core domain which are involved in binding and sliding on unfolded OMPs. At the beginning of the manuscript they also apply single-molecule FRET spectroscopy, which yields to them that the core domain alone binds to unfolded OmpX and they observe two very distinct states in OmpX, suggesting that OmpX exists in solution and in complex in two conformations.

Overall, the NMR data appear very convincing of the manuscript and I have only minor questions. In particular, I find the b-strand peptide approach very creative. The FRET experiments, however, raise a few questions that should be addressed. Overall the study stands very strong with the NMR data and it has not become very clear to me what new insight compared to an earlier study the FRET experiments yield, besides the SurA core-mediated expansion of OmpX.

- The authors report an earlier study of OmpX in complex with SurA. In this earlier study, unfolded OmpX was not described as a two-state model but apparently as a board ensemble of multiple states. While discrepancy is fine, I was wondering what the second, apparently well-defined state of unfolded OmpX is. Can the authors speculate what a distinct sharp state could be? For an unstructured, hydrophobic polypeptide I would agree that multiple states are likely to expect, but would wonder if it is only two states.

Additionally, the earlier study pointed out that fluorophores are transiently interacting with the unfolded OmpX, therefore I was wondering if the authors of this manuscript tested anisotropy and if the experiments were performed at room temperature or elevated temperature.

- I was wondering what justifies the assumption that the two OmpX states are the same in absence and presence of SurA. This is at least suggested by Figure 2.

- The earlier FRET study also reported binding of multiple SurA to OmpX. Was this also observed here, or can it be excluded? Could the second distinct state be a second SurA binding and sliding?

- The earlier FRET study reported incomplete binding even at $> 2 \mu\text{M}$ SurA. Do the authors observe similar behaviour? In particular, the FRET efficiency distribution coloured in red in Fig 2b and 2c show both a very broad tail to high FRET efficiency, nearly appearing like a hump. This could be still uncomplexed unfolded OmpX, which would suggest that the kinetic scheme and the underlying Markov Model is incorrect. Please justify. Can this be verified by a three-color FRET experiment ensuring colocalization and species filtering?

- single-molecule FRET experiments were performed in presence of 0.02% Tween-20, not far from the cmc. How does Tween-20 interact or affect unfolded OmpX structure? Does it induce partial folding and a second distinct FRET state?

- The NMR experiments were performed in presence of 0.8M urea. How does this affect SurA

structure, stability, dynamics?

- How do the authors ensure that unfolded OMPs do not aggregate at the high concentrations required for NMR experiments?

- I very much like the peptide approach. Why did the authors only use 15mer peptides? In particular their finding that the peptide affinity is so much lower than OmpX or other OMPs as well as that the 15mer peptides apparently bind only one of the two binding regions suggests that a longer peptide could bridge. Such an experiment could be very meaningful to understand the affinity differences from μM to $100 \mu\text{M}$.

- I am puzzled by the Fig 2 model vs the Fig 6 model. Fig 2 suggests that OMPs could be in two distinct states (compact and expanded) on its own and in complex with SurA. Fig 6 suggests an expanded uOMP in complex with SurA and a compact uOMP in unbound state, something which agrees well with earlier studies.

smaller questions:

- A minor note: several earlier studies, which are also referenced by the authors already suggested that the expansion might be a way to present a linear OMP to BAM. The discussion and Fig 6 legend suggest that this is a 'new' finding of this study here, though I would see it even further supporting the earlier findings and suggestions. Maybe rephrase or point to earlier studies?

- can the authors please report full protein sequences of the OMPs and truncated OMPs in the SI? Where is the Cysteine in tBamA?

- I found a reported K_i instead of K_d on page 7.

- sometimes brackets were missing

Reviewer #2 (Remarks to the Author):

In this study, Schiffrin et al. employ smFRET and methyl-TROSY NMR to identify the sites within the periplasmic chaperone SurA that bind to a model E. coli outer membrane protein (OMP) called OmpX and to determine the effects of the interaction between SurA and OmpX on the conformation of both binding partners. The use of NMR to investigate the binding sites on SurA is commendable because the method is fundamentally less biased than cross-linking or smFRET, which typically focus on the relative positions of only a small number of amino acids in two interacting proteins. While the quality of the data is high and the study presents valuable insights into the location of the SurA binding sites and the conformational changes that both SurA and OmpX undergo upon substrate binding, several concerns need to be addressed to strengthen the scientific rigor of the manuscript. Furthermore, as noted below, many of the results are not especially novel. For this reason, the manuscript in its current form does not significantly advance our understanding of SurA function.

Major comments:

1. The evolutionary conservation of PPLase domains in SurA described in Fig. 1c was previously investigated by some of the same authors and led to a similar conclusion (ref. 49).
2. Previous studies (refs. 62 and 63) have examined the expansion of client OMPs bound to SurA using smFRET. Fig. 2 does not provide significant new insights because the earlier studies led to similar conclusions.
3. Previous studies have mapped the substrate binding sites within SurA through cross-linking (refs. 54 and 56) and NMR (ref. 55), and have led to conclusions that are similar to those based on the results shown in Fig. 3 (i.e., that substrates bind to the core and P1 domains of SurA).
4. Previous investigations have analyzed the conformational changes that SurA undergoes when bound to client OMPs or OMP peptides using NMR (ref. 55) and smFRET (refs. 55 and 56). The results have led to conclusions that are similar to those based on the data shown in Figs. 4 and 5.
5. In the smFRET experiments, if alternating laser excitation (ALEX) or pulsed interleaved excitation (PIE) techniques were utilized, these methods should be explicitly mentioned in the main text and described in detail in the Methods section.
6. While the application of multi-parameter photon-by-photon hidden Markov modeling (mpH2MM) to extract the rate of conformational change is noteworthy, limitations arise due to the confined detection window imposed by diffusion. The authors can only see events that occur within very short timescales (< 1 ms, when the molecules are still in the confocal volume), but most of the dynamics they observed occurred on a time scale that was longer than 1 ms (Table S1). The interpretation of the data may not be accurate because the slower conformational changes may have been influenced by the rate of diffusion.
7. It has previously been observed that the smFRET distributions of apo-OMPs in aqueous buffers without denaturants are broad, indicating the existence of a heterogeneous ensemble of conformations. It is therefore likely that apo-OmpX exists in more than two conformational states (apo-expanded and apo-compact). In this study the higher FRET population of SurA-WT and SurA-core bound OmpX (in which E spans from 0.4 to 1.0) is very similar to the entire FRET distribution of apo-OmpX. Thus in the presence of SurA, OmpX may not exist in only bound-expanded and bound-compact states, as the authors suggest. An alternative explanation of the data is that the low FRET values represent a bound state and the high FRET values (which are the same observed for apo-OmpX) represent an unbound state. To distinguish between these two possibilities, the authors should vary the SurA concentration to see if they observe changes in the ratio of the two populations.
8. The argument that OmpX exists in bound-expanded and bound-compact states is based on smFRET experiments in which the distances between the two ends of the protein are measured. Given that the location of the internal portions of OmpX cannot be determined from the data, what is the physiological significance of the two states? The authors should explain how one of the two conformations prevents the protein from aggregating and promotes delivery to BAM.

9. The deletion of P1 and P2 did not alter the FRET distribution of each state (Fig. 2b-c) but significantly increased the fraction of expanded OmpX (Fig. 2e). These results suggest that the core domain expands unfolded OmpX while the PPIase domains compact OmpX. However, the authors demonstrated that P1 could bind to an OmpX peptide (Fig. 3), potentially leading to further expansion of OmpX. The authors should provide an explanation for this discrepancy.

10. The authors should explain why they chose to label isoleucine and methionine residues in OmpX in their NMR experiments? Could this selection introduce any bias?

11. In ref. 56, four lysine residues in the P2 domain were found to cross-link to OmpX. However, in the present work, no interaction was observed between OmpX and the P2 domain. The authors should provide an explanation for this disparity.

12. In ref. 49, it was demonstrated that SurA Δ P2 exhibits significantly lower binding affinity to OmpT and fails to prevent its aggregation, suggesting a binding interaction between P2 and OmpT. To determine if there are significant differences in the sites within SurA that bind different OMPs, the authors should compare OmpT to OmpX in the presence of SurA using NMR to see if they can detect any interactions between P2 and OmpT.

13. In the NMR experiments shown in Fig. 3, specific residues located between the two 'lobes' of the SurA N-domain (residues I91, M109, M114, I132, and M136), as well as residue M231 in the P1 domain, are significantly affected when bound to either OmpX or OmpX-derived peptides. The data suggest that the residues form a functional protein-protein interaction site. To test this possibility more rigorously, the authors should consider mutating some of these residues and assessing the impact of the mutations on binding.

14. Because the size and sequence of OMPs differ considerably, the number and location of potential SurA binding sites must also differ. Nevertheless, SurA appears to play a major role in preventing the aggregation of most OMPs and promoting their delivery to BAM. The authors should provide an explanation for the ability of SurA to function as a chaperone for such a diverse array of proteins.

Minor comments:

1. Please define SANS and smFRET in the middle of p.4.

2. The authors should explain more clearly how the results of TROSY NMR are interpreted (e.g., the significance of a peak that is not overlapped or broadened or has low intensity) to cater to the broad readership of Nature Communications, some of whom may be unfamiliar with NMR techniques.

3. On the bottom of p. 6, the authors state that "Of the 18 peaks corresponding to residues in the core domain, 12 were severely affected with intensity ratios close to zero, while the intensities of the remaining 6 core peaks were ~50% reduced." However, in Fig. 3a there are only 8 peaks (marked

by stars) but not 12 with intensity ratios close to zero in the core domain.

4. In the first paragraph on p. 7, “. consistent with previous cross-linking MS and HDX-MS data.” should be “, which is consistent with previous cross-linking MS and HDX-MS data.”

5. In the second paragraph on p. 7, “These clients bind SurA-WT with K_d 3.7 ± 1.0 μ M, K_i 5.2 ± 1.7 μ M, and K_d 1.4 ± 0.1 μ M for tOmpA, OmpF and tBamA...” Is K_i simply a typo? If not, K_i should be explained.

5. The SurA-WT + OmpX data in Fig. 3a-b and in Fig. S10 are slightly different. Is this simply an error? If not, an explanation of the disparity should be provided.

Reviewer #3 (Remarks to the Author):

The authors provide a comprehensive study for the interaction of the periplasmatic chaperone SurA and outer membrane proteins. Understanding chaperoning at molecular level requires a detailed picture of the protein binding specificity. So far this is poorly understood for SurA. The present study provides the urgently needed molecular detail. Technically, this is an achievement as the client group is particularly aggregation-prone. The authors used in particular high resolution NMR and sm-FRET to investigate this, and this is very well done and extensively (some readers may feel even "excessively") documented. A particularly interesting aspect of the study is that SurA is an ATP-independent chaperone, and substrate interaction of such chaperones is far less understood than that of ATP-dependent chaperone machines.

Minor comments:

1. The authors should also discuss more clearly the options for client release. This is an interesting aspect given that it is an ATP-independent chaperone.

2. Fig. S11 - location of OmpX-derived peptides

a. This figure should be in the main part of the paper, as the data on the peptides cannot be understood without this information.

b. Please list the exact boundaries of each of the peptides.

3. Fig. S15 - conservation

This figure visualises the conserved regions in the structure. A sequence alignment should be added in the figure, indicating the highlighted regions in the protein sequence.

Reviewer Comments (NCOMMS-24-03929)

Reviewer #1 (Remarks to the Author):

Schiffirin, Crossley and colleagues present in this manuscript a study of the periplasmic chaperone SurA. In particular they report an extensive study of potential binding sites of unfolded outer membrane proteins and peptides to SurA using NMR spectroscopy. They reveal various important residues in SurA in P1 and the core domain which are involved in binding and sliding on unfolded OMPs. At the beginning of the manuscript they also apply single-molecule FRET spectroscopy, which yields to them that the core domain alone binds to unfolded OmpX and they observe two very distinct states in OmpX, suggesting that OmpX exists in solution and in complex in two conformations.

Overall, the NMR data appear very convincing of the manuscript and I have only minor questions. In particular, I find the b-strand peptide approach very creative. The FRET experiments, however, raise a few questions that should be addressed. Overall the study stands very strong with the NMR data and it has not become very clear to me what new insight compared to an earlier study the FRET experiments yield, besides the SurA core-mediated expansion of OmpX.

- The authors report an earlier study of OmpX in complex with SurA. In this earlier study, unfolded OmpX was not described as a two-state model but apparently as a board ensemble of multiple states. While discrepancy is fine, I was wondering what the second, apparently well-defined state of unfolded OmpX is. Can the authors speculate what a distinct sharp state could be? For an unstructured, hydrophobic polypeptide I would agree that multiple states are likely to expect, but would wonder if it is only two states.

We thank the Reviewer for highlighting that we need to be clearer when describing the FRET states we report, and the potential multitude of conformational states reported previously by Chamachi et al. There is actually no discrepancy: it reflects differences in the type of smFRET experiments we have each used, that are complimentary. Our work focuses on fluctuations on the millisecond timescale (~0.1 - 100 ms), whilst Chamachi et al. used methods that sample (and average) changes on a broader range of timescales. Most importantly, we concur with the referee that the two conformations we detect will not describe the entire energy landscape of the unfolded polypeptide, which will presumably explore conformational rearrangements on faster (sub microsecond) and longer (sec to minute) timescales, which would not be detected by our method of analysis. The reason we chose to focus on rapid fluctuations on the msec timescale is related to the inter-domain dynamics of SurA which have been shown to occur on a msec timescale, both intrinsically and when bound to OmpX¹.

We have amended the legend to Fig. 2 to clarify that we expect a broad range of conformers within each FRET state, as well as adding the justification of chosen timescales to the main text.

Additionally, the earlier study pointed out that fluorophores are transiently interacting with the unfolded OmpX, therefore I was wondering if the authors of this manuscript tested anisotropy and if the experiments were performed at room temperature or elevated temperature.

This a good point. We have now performed the requested anisotropy experiments. The results (presented as new Supplementary Table S1) show that the dyes are freely rotating. We also now mention these new experiments in the main text and Supplementary Methods sections. Notably, similar experiments were performed by Chamachi et al. who also reported similar values (see Table S1 in their manuscript).

Our smFRET experiments were conducted at room temperature. We have now added this to the main text Methods section 'Data Acquisition'.

- I was wondering what justifies the assumption that the two OmpX states are the same in absence and presence of SurA. This is at least suggested by Figure 2.

We apologise if this was not clear. In fact, the FRET states for OmpX in the absence and presence of SurA are not the same (denoted differently as apo-expanded/apo-compact and bound-expanded/bound-compact in Fig. 2) as evidenced by the difference in their E_{raw} values (now additionally quoted in the legend to Fig. 2). We have amended the legend to Fig. 2 to make this clear.

- The earlier FRET study also reported binding of multiple SurA to OmpX. Was this also observed here, or can it be excluded? Could the second distinct state be a second SurA binding and sliding?

There is no way to tell from our single-molecule data if multiple SurA molecules are bound to OmpX and this is a possibility (and could contribute to the dynamics and expansion we observe). We have now added to the main text to highlight such a possibility.

- The earlier FRET study reported incomplete binding even at $> 2 \mu\text{M}$ SurA. Do the authors observe similar behaviour? In particular, the FRET efficiency distribution coloured in red in Fig 2b and 2c show both a very broad tail to high FRET efficiency, nearly appearing like a hump. This could be still uncomplexed unfolded OmpX, which would suggest that the kinetic scheme and the underlying Markov Model is incorrect. Please justify. Can this be verified by a three-color FRET experiment ensuring colocalization and species filtering?

Our smFRET data were collected using SurA-WT and SurA-core concentrations of $30 \mu\text{M}$ so that OmpX is $>95\%$ saturated with the chaperone, based on the k_D presented in Chamachi et al. ($\sim 1.0 \mu\text{M}$) and our previous work¹. These data are fully consistent with those in Chamachi et al. who also show saturation with OmpX at SurA concentrations of $12 \mu\text{M}$. Hence the peak the referee point to cannot arise from unbound OmpX. A three colour FRET experiment using colocalization to identify binding events as Reviewer #1 suggests is not possible due to the μM concentrations of SurA needed for binding in comparison to the pM needed for single-molecule detection.

- single-molecule FRET experiments were performed in presence of 0.02% Tween-20, not far from the cmc. How does Tween-20 interact or affect unfolded OmpX structure? Does it induce partial folding and a second distinct FRET state?

The presence of 0.02% (v/v) Tween-20 does not affect our results. To show this we collected data without Tween-20 and applied the same analysis method described in our manuscript to derive the best fit number of states. As shown below, the presence of 0,02% (v/v) Tween-20 made no difference to the number of states needed to best describe the data.

Small concentrations of Tween-20 are widely used in smFRET studies, including on OMPs². It is important also to note that while in the experiments reported in Chamachi et al. detergent is not added, there is still detergent present in the experiment due to the dilution protocol used (in our experiments we dilute from a stock containing Gdn-HCl). Addition of Tween-20 was used to prevent non-specific adsorption to the glass interface allowing enough single-molecule bursts to be collected to accurately quantify dynamics. We have now edited our methods to explain the use of Tween-20.

- The NMR experiments were performed in presence of 0.8M urea. How does this affect SurA structure, stability, dynamics?

We have previously shown that SurA-WT is stably folded in 0.8M urea (it does not begin to unfold until >3 M urea³). Accordingly, the presence of 0.8 M urea has little effect on the methyl-TROSY spectrum of SurA-WT as shown below (Fig. 1).

Importantly, for the experiments involving OMP binding in which 0.8 M urea is present, the reference spectrum for SurA-WT alone also contains 0.8 M urea. Hence all changes are due to the presence of the OMP, and not the urea.

Fig. 1. Comparison of Ile and Met peak intensity ratios (left) and chemical shift perturbations (right) between SurA-WT in the presence or absence of 0.8 M urea. Samples contained 5 μ M SurA \pm 0.8 M urea, 5 mM EDTA, 20 mM Tris-HCl, pH 8, at 25 $^{\circ}$ C.

- How do the authors ensure that unfolds OMPs do not aggregate at the high concentrations required for NMR experiments?

Careful selection of buffer conditions (urea concentration, pH, salt) and concentrations of SurA and OMPs were key to the success of these experiments. A pH of no lower than 8.0, the exclusion of added salt, low micromolar OMP concentrations (5 μ M in our experiments), and a concentration of urea of \sim 0.8 M are all important parameters in minimising OMP self-association and aggregation in the absence of chaperones^{4,5}. For example, the Fleming group has demonstrated that unfolded OmpX is monomeric at \sim 9 μ M at pH 8 in the absence of salt at 1M urea⁴. In our experience, the concentration of urea is particularly important parameter in maintaining OMP solubility. In the present work, the concentration of SurA (5 μ M) was also carefully selected to allow a substantial fraction bound (\sim 40-60%) of both the OMP (to additionally disfavour aggregation) and SurA (for detecting changes upon binding). The presence of SurA at this concentration also enabled sufficient signal-to-noise in the methyl-TROSY experiments to be achieved within a few hours (\sim 5h), to minimise the potential for aggregation over extended time periods.

- I very much like the peptide approach. Why did the authors only use 15mer peptides? In particular their finding that the peptide affinity is so much lower than OmpX or other OMPs as well as that the 15mer peptides apparently bind only one of the two binding regions suggests that a longer peptide could bridge. Such an experiment could be very meaningful to understand the affinity differences from μ M to 100 μ M.

The lengths of the peptides used were chosen as the approximate lengths of the β -strands in the folded state of OmpX. We agree with the reviewer that experiments varying the lengths of peptides to see how multiple binding sites affect the affinity would be an excellent approach. Other experiments such as truncating peptides, varying the position of the binding motifs along the chain, and mutagenesis of residues within and around the aromatic-rich binding motifs will also likely be very informative. We see all these as part of a much more detailed investigation into the sequence and length determinants of SurA-client interactions using OMP-derived peptides. However, such a study is a large undertaking and will be the focus of our future work.

- I am puzzled by the Fig 2 model vs the Fig 6 model. Fig 2 suggests that OMPs could be in two distinct states (compact and expanded) on its own and in complex with SurA. Fig 6 suggests an expanded uOMP in complex with SurA and a compact uOMP in unbound state, something which agrees well with earlier studies.

We agree that this was confusing, and we have edited Fig. 6 to improve its clarity and to better link to the smFRET data shown in Fig. 2. We hope you agree this is now improved.

Smaller questions:

- A minor note: several earlier studies, which are also referenced by the authors already suggested that the expansion might be a way to present a linear OMP to BAM. The discussion and Fig 6 legend suggest that this is a 'new' finding of this study here, though I would see it even further supporting the earlier findings and suggestions. Maybe rephrase or point to earlier studies?

We thank the reviewer for pointing out this omission. We have now added these references to the legend of Fig. 6 (Marx et al.⁶ and Chamanchi et al.⁷) where these ideas are discussed.

- can the authors please report full protein sequences of the OMPs and truncated OMPs in the SI? Where is the Cysteine in tBamA?

We have added a table to the Supplementary Data file which gives the full sequences of OMPs (and SurA variants) used in the study, and added a reference to this in the Methods section. Details of the Cys-tBamA plasmid construct are provided in the Methods which we reproduce below for reference:

“The Cys-tBamA construct for use in microscale thermophoresis (MST) experiments, was also obtained from TWIST bioscience in a pET29b+ expression vector. This construct contained the N-terminal sequence Met-Gly-Ser-Cys followed by the codon-optimised mature tBamA sequence (residues 425-810). The two natural Cys residues in tBamA were mutated to Ser (C690S/C700S).”

- I found a reported K_i instead of K_d on page 7.

K_i rather than K_d is used here for the binding of unfolded OmpF to SurA as this number reflects an inhibition constant (K_i) obtained from phage-based ELISA assays in Bitto et al (2004)⁸. We thank the reviewer for this raising this, and we have added a sentence to explain this in the main text.

- sometimes brackets were missing

We have checked and corrected these.

Reviewer #2 (Remarks to the Author):

In this study, Schiffrin et al. employ smFRET and methyl-TROSY NMR to identify the sites within the periplasmic chaperone SurA that bind to a model E. coli outer membrane protein (OMP) called OmpX and to determine the effects of the interaction between SurA and OmpX on the conformation of both binding partners. The use of NMR to investigate the binding sites on SurA is commendable because the method is fundamentally less biased than cross-linking or smFRET, which typically focus on the relative positions of only a small number of amino acids in two interacting proteins. While the quality of the data is high and the study presents valuable insights into the location of the SurA binding sites and the conformational changes that both SurA and OmpX undergo upon substrate binding, several concerns need to be addressed to strengthen the scientific rigor of the manuscript. Furthermore, as noted below, many of the results are not especially novel. For this reason, the manuscript in its current form does not significantly advance our understanding of SurA function.

Major comments:

1. The evolutionary conservation of PPLase domains in SurA described in Fig. 1c was previously investigated by some of the same authors and led to a similar conclusion (ref. 49).

We previously reported the evolution in conservation of PPIase domains in SurA in our 2019 paper⁹. This analysis focussed on a relatively small number of homologues (~1000 from proteobacteria) to investigate possible correlations between OMP properties and SurA domain architecture in different proteobacterial classes. Here, we have analysed a much larger number of homologues from the INTERPRO database (~14,000), and we can now take advantage of AlphaFold2, which was not available when we conducted our original study, to predict and analyse the structures of these homologues, rather than rely on pfam predictions of domain architectures. Our data reveal that the majority of 'core alone' homologues in the pfam database are actually SurA homologues containing PPIase domains which have not been successfully predicted to be present. In our previous manuscript we stated that our data: "suggests that the ancestral SurA chaperone comprised either the core domain alone, as previously proposed [24], or the core domain with one PPIase domain"⁹. Our new data makes it clear that the ancestral chaperone most likely contained the core domain and one PPIase domain, and that the loss of all PPIase domains only occurs in very rare cases in organisms living in highly specialised environments. This is important in the context of the current paper as we have uncovered new roles for PPIase domains in the modulation of OMP conformational dynamics, as well as demonstrating that OMP substrates bind to the P1 domain.

2. Previous studies (refs. 62 and 63) have examined the expansion of client OMPs bound to SurA using smFRET. Fig. 2 does not provide significant new insights because the earlier studies led to similar conclusions.

We are aware of these previous smFRET studies which also observed expansion of OMP substrates in the presence of SurA, and we cite the relevant references in the main text of our manuscript. Here, we have expanded on these results, using smFRET combined with Hidden Markov Modelling of the resulting data to determine how the SurA domains modulate substrate dynamics. Our results can be summarised in two points. We show that:

- SurA can expand unfolded OMPs using its core domain alone.
- Substrate expansion is modulated by SurA PPIase domains which limit the extent of expansion and do so by altering the rates of OMP chain dynamics.

Both points provide key insights into how SurA functions, especially in the context of our finding that the presence of at least one PPIase domains SurA is so highly conserved.

We hope that these points are clear in our manuscript, and we have checked the text to ensure this is the case.

3. Previous studies have mapped the substrate binding sites within SurA through cross-linking (refs. 54 and 56) and NMR (ref. 55), and have led to conclusions that are similar to those based on the results shown in Fig. 3 (i.e., that substrates bind to the core and P1 domains of SurA).

Again, we are aware of these previous studies, which we cite in several places in our manuscript. But again, the important point is that the results we present provide new and much more detailed information than was known hitherto. In Calabrese et al. (ref 56) the crosslinking data suggested interactions across the whole surface of the core domain, as well as to P1. In Marx et al. (ref 54) cross-links were additionally detected to the P2 domain. However, there are difficulties with cross-

linking approaches, including the potential to overemphasise very lowly populated species (long timescales of crosslinking are required), a lack of atomic-level detail (residues that can be detected are limited by the crosslinking chemistry used), and an inability to detect SurA conformational changes associated with substrate binding. In reference 55 (Jia et al, 2020¹⁰) no OMP substrate binding experiments were performed.

A key novelty of our approach is that not only do we detect the 'diffuse' binding of the OMP chain on the chaperone surface (in particular, across the core domain), but we have also been able to identify the binding 'hotspot' regions which mediate the affinity between OMPs and SurA. The identification of these regions is of crucial importance for precisely targeting SurA with inhibitory molecules and presents new insights into SurA-client recognition that was not known from these previous reports.

4. Previous investigations have analyzed the conformational changes that SurA undergoes when bound to client OMPs or OMP peptides using NMR (ref. 55) and smFRET (refs. 55 and 56). The results have led to conclusions that are similar to those based on the data shown in Figs. 4 and 5.

The reviewer is correct in that both the binding of WEYIPNV and the SurA-S220A mutation are known to promote a conformational change favouring core-P1 open states. Here, we show with atomic-level detail that the conformational ensembles induced by WEYIPNV binding and SurA-S220A are extremely similar (**Fig. S17**). By identifying the NMR 'fingerprint' for this conformational change, we are then able to use the information to demonstrate that there are specific signals within OMP sequences that promote this conformational change (**Fig. S16**). As P1-Core opening is known to generate a more 'active' chaperone (Soltes et al., 2016¹¹), this indicates a novel activation mechanism for an ATP-independent chaperone in which, binding of the substrate itself promotes the movement of folded domains to activate the chaperone. This contrasts with the mechanisms of activation of other ATP-independent chaperones that utilise oligomerisation or an order-to-disorder transition to activate the chaperone state. We make this clear in the Discussion of our manuscript.

5. In the smFRET experiments, if alternating laser excitation (ALEX) or pulsed interleaved excitation (PIE) techniques were utilized, these methods should be explicitly mentioned in the main text and described in detail in the Methods section.

We have now detailed that we used pulsed interleaved excitation in the main text and added another reference to this in the Supplementary Methods section where we describe in detail the equipment used.

6. While the application of multi-parameter photon-by-photon hidden Markov modeling (mpH2MM) to extract the rate of conformational change is noteworthy, limitations arise due to the confined detection window imposed by diffusion. The authors can only see events that occur within very short timescales (< 1 ms, when the molecules are still in the confocal volume), but most of the dynamics they observed occurred on a time scale that was longer than 1 ms (Table S1). The interpretation of the data may not be accurate because the slower conformational changes may have been influenced by the rate of diffusion.

The initial publication describing mpH²MM analysis quantifies the dynamics in a DNA hairpin which exhibits slower dynamics (~6 ms) than we present here¹². The first examples of hidden Markov modelling (HMM) on diffusion-based FRET data showed that recovering rates between ~10 μ s and ~1 s is achievable¹³. It is important to note that the ability to measure dynamics longer than the diffusion time is not limited to HMM and that examples of similar commonly used smFRET analysis techniques which show examples of dynamics slower than the diffusion time have been reported¹⁴⁻¹⁷, including in an excellent and comprehensive review covering the topic of analysis methods in smFRET and the rates they can recover¹⁸. Recovering the rates of dynamic interchange using analysis of diffusion-based smFRET data is **not** strictly limited by the diffusion time of the molecule and it is possible to measure rates slower (~1000x slower using HMM) than the average detection window.

We thank the Reviewer #2 for highlighting this important point that could be missed by the readers of our manuscript. We now specify the timescales it is possible to detect using our chosen methods and we have amended the Methods section describing our data analysis to clarify this point, and have added citations to the appropriate references.

7. It has previously been observed that the smFRET distributions of apo-OMPs in aqueous buffers without denaturants are broad, indicating the existence of a heterogeneous ensemble of conformations. It is therefore likely that apo-OmpX exists in more than two conformational states (apo-expanded and apo-compact). In this study the higher FRET population of SurA-WT and SurA-core bound OmpX (in which E spans from 0.4 to 1.0) is very similar to the entire FRET distribution of apo-OmpX. Thus in the presence of SurA, OmpX may not exist in only bound-expanded and bound-compact states, as the authors suggest. An alternative explanation of the data is that the low FRET values represent a bound state and the high FRET values (which are the same observed for apo-OmpX) represent an unbound state. To distinguish between these two possibilities, the authors should vary the SurA concentration to see if they observe changes in the ratio of the two populations.

Given the scenario presented, OmpX would have to be ~50% unbound at the concentrations used, which given the known K_D is impossible (see our response to point 5 of Reviewer #1). While the FRET values do span similar regions it can be seen clearly that they are **not** the same (see Fig. below). Therefore, the OmpX SurA-WT bound-compact FRET state could not be made up of unbound OmpX.

8. The argument that OmpX exists in bound-expanded and bound-compact states is based on smFRET experiments in which the distances between the two ends of the protein are measured. Given that the location of the internal portions of OmpX cannot be determined from the data, what is the physiological significance of the two states? The authors should explain how one of the two conformations prevents the protein from aggregating and promotes delivery to BAM.

These questions raised by Reviewer #2 are interesting points that we can only currently speculate on. How and why the conformational ensemble affects aggregation prevention and delivery to BAM is not clear. Answering these questions will require more detailed analysis with FRET dyes in different places on a variety of OMP sequences, and combined with careful analysis of the concentration dependence of aggregation with/without SurA (and domain deletion variants), and detailed analysis of the rates of folding of different sequences via BAM. This is way beyond the scope of our current manuscript, but will build on it, and will be the focus of our future work.

9. The deletion of P1 and P2 did not alter the FRET distribution of each state (Fig. 2b-c) but significantly increased the fraction of expanded OmpX (Fig. 2e). These results suggest that the core domain expands unfolded OmpX while the PPIase domains compact OmpX. However, the authors demonstrated that P1 could bind to an OmpX peptide (Fig. 3), potentially leading to further expansion of OmpX. The authors should provide an explanation for this discrepancy.

We thank Reviewer #3 for highlighting a typographical error we made in the Discussion section which led to this apparent discrepancy in our manuscript. We intended to write “...two binding hotspots could facilitate chain **dynamics** in the bound state” not “chain expansion” as was written.

10. The authors should explain why they chose to label isoleucine and methionine residues in OmpX in their NMR experiments? Could this selection introduce any bias?

In all our NMR experiments it is the SurA chaperone rather than its substrates, such as OmpX, that is isotopically labelled. Isoleucine and methionine were selected as they provide excellent coverage for methyl-directed NMR (methyl-TROSY) in all regions of the chaperone surface (as illustrated in **Fig. 5a**), thus minimising the potential for bias in the detection of sites of substrate interaction.

11. In ref. 56, four lysine residues in the P2 domain were found to cross-link to OmpX. However, in the present work, no interaction was observed between OmpX and the P2 domain. The authors should provide an explanation for this disparity.

The cross-links detected between the SurA P2 domain and OmpX in Calabrese et al. (ref 56)¹ were obtained using DSBU as the cross-linking reagent. DSBU has been shown to be able to cross-link residues separated by a straight-line distance of ~27-30 Å¹⁹, therefore these cross-links are not necessarily indicative of direct interaction. Cross-linking MS is a highly sensitive, irreversible technique that can detect very weak, lowly populated interactions and also requires long incubation times (e.g. 45 minutes was used in ref 56). Our NMR experiments require a population of ~5-10% to be measurable, so we cannot rule out very weak, lowly populated, or very transient interactions between OmpX and P2. However, our data clearly indicate that for the OMPs tested this domain is not one of the main interaction sites. We hope this is clear (see **Fig. S10** in our revised manuscript).

12. In ref. 49, it was demonstrated that SurA ΔP2 exhibits significantly lower binding affinity to OmpT and fails to prevent its aggregation, suggesting a binding interaction between P2 and OmpT. To determine if there are significant differences in the sites within SurA that bind different OMPs, the authors should compare OmpT to OmpX in the presence of SurA using NMR to see if they can detect any interactions between P2 and OmpT.

We did not select OmpT for our study for practical reasons: it readily oligomerises at the low μM concentrations employed needed, and previous work using analytical AUC demonstrated that of a panel of 8 OMPs, OmpT is the most prone to self-associate under similar conditions to those used in our NMR experiments⁴. Nevertheless, we have now attempted to detect OmpT-SurA interactions by NMR, as the referee suggested, using OmpT concentrations of 5 μM and 15 μM, but did not observe any interactions, presumably because of its self-association. Our data cannot rule out that P2 isn't important for substrate binding in the case of some OMPs. *In vivo* data also suggest that P2 is involved in the relief of the autoinhibitory effect of P1 on the core domain¹¹. Therefore, in the specific case of SurA ΔP2 being unable to prevent the aggregation of OmpT, it is possible that this is due to the inhibition of chaperone activity by P1 in the absence of P2. We have added a comment on the potential role of P2 in relieving the auto-inhibition of the core domain by P1 in our revised manuscript.

13. In the NMR experiments shown in Fig. 3, specific residues located between the two 'lobes' of the SurA N-domain (residues I91, M109, M114, I132, and M136), as well as residue M231 in the P1 domain, are significantly affected when bound to either OmpX or OmpX-derived peptides. The data suggest that the residues form a functional protein-protein interaction site. To test this possibility more rigorously, the authors should consider mutating some of these residues and assessing the impact of the mutations on binding.

We agree with the reviewer that mutational studies to determine the relative importance of different areas of the binding hotspots identified for different OMP peptides and OMP sequences will be fascinating to perform. The sites are large containing many residues, and so we see this a whole new project to pursue in the future (see point 14 below).

14. Because the size and sequence of OMPs differ considerably, the number and location of potential SurA binding sites must also differ. Nevertheless, SurA appears to play a major role in preventing the aggregation of most OMPs and promoting their delivery to BAM. The authors should provide an explanation for the ability of SurA to function as a chaperone for such a diverse array of proteins.

Good point. We believe the specificity of SurA for aromatic-rich motifs, and in particular Ar-X-Ar motifs which are common in OMPs (where they often form part of the 'aromatic girdle' at the membrane interface regions), helps explain how the chaperone can bind many different OMPs which exhibit large diversity of sequence, size, and structures. We have now added a comment about this to the Discussion.

Minor comments:

1. Please define SANS and smFRET in the middle of p.4.

Thank you - these have been added.

2. The authors should explain more clearly how the results of TROSY NMR are interpreted (e.g., the significance of a peak that is not overlapped or broadened or has low intensity) to cater to the broad readership of Nature Communications, some of whom may be unfamiliar with NMR techniques.

We have added a section to page 6, explaining that the two different types of peak changes (shifts and/or broadening) observed in methyl-TROSY NMR report on binding or conformational change at the location of the corresponding residue. We also have added references to a useful paper on the interpretation of 2D NMR spectra for readers not familiar with the technique^{20,21}.

3. On the bottom of p. 6, the authors state that "Of the 18 peaks corresponding to residues in the core domain, 12 were severely affected with intensity ratios close to zero, while the intensities of the remaining 6 core peaks were ~50% reduced." However, in Fig. 3a there are only 8 peaks (marked by stars) but not 12 with intensity ratios close to zero in the core domain.

We have reworded this to add clarity: "Of the 18 peaks corresponding to residues in the core domain, 9 were severely affected with intensity ratios close to zero (indicated by asterisks in **Fig. 3c**), 2 were ~80% reduced, while the intensities of the remaining 7 core peaks were ~50% reduced"

4. In the first paragraph on p. 7, ". consistent with previous cross-linking MS and HDX-MS data." should be ", which is consistent with previous cross-linking MS and HDX-MS data."

We have amended the text as suggested by the reviewer.

5. In the second paragraph on p. 7, “These clients bind SurA-WT with K_d $3.7 \pm 1.0 \mu\text{M}$, K_i $5.2 \pm 1.7 \mu\text{M}$, and K_d $1.4 \pm 0.1 \mu\text{M}$ for tOmpA, OmpF and tBamA...” Is K_i simply a typo? If not, K_i should be explained.

K_i rather than K_d is used here for the binding of unfolded OmpF to SurA as this number reflects an inhibition constant (K_i) obtained from phage-based ELISA assays in Bitto et al (2004)⁸. We thank the reviewer for this raising this and have added a sentence to explain this in the main text.

6. The SurA-WT + OmpX data in Fig. 3a-b and in Fig. S10 are slightly different. Is this simply an error? If not, an explanation of the disparity should be provided.

We thank the reviewer for their careful examination of the figures. This was an error in that the wrong spectrum was used, which we have now corrected.

Reviewer #3 (Remarks to the Author):

The authors provide a comprehensive study for the interaction of the periplasmic chaperone SurA and outer membrane proteins. Understanding chaperoning at molecular level requires a detailed picture of the protein binding specificity. So far this is poorly understood for SurA. The present study provides the urgently needed molecular detail. Technically, this is an achievement as the client group is particularly aggregation-prone. The authors used in particular high resolution NMR and sm-FRET to investigate this, and this is very well done and extensively (some readers may feel even "excessively") documented. A particularly interesting aspect of the study is that SurA is an ATP-independent chaperone, and substrate interaction of such chaperones is far less understood than that of ATP-depending chaperone machines.

Minor comments:

1. The authors should also discuss more clearly the options for client release. This is an interesting aspect given that it is an ATP-independent chaperone.

We have added a section to the Discussion on the open question of how client release for folding into the outer membrane could be achieved.

2. Fig. S11 - location of OmpX-derived peptides

a. This figure should be in the main part of the paper, as the data on the peptides cannot be understood without this information.

We agree. We have added it to Figure 3 (a,b), and removed Fig S11.

b. Please list the exact boundaries of each of the peptides.

The list of exact boundaries for each peptide can be found in Supplementary Table S3 (previously Table S2).

3. Fig. S15 - conservation

This figure visualises the conserved regions in the structure. A sequence alignment should be added in the figure, indicating the highlighted regions in the protein sequence.

We agree and thank the reviewer for this suggestion. We have added a new part (d) to this Supplementary Fig.S15 which indicates the highly conserved residues from the alignment of three domain SurA homologues on the protein's sequence, coloured as in the visualisation in part (c).

References in Response to Referees above

1. Calabrese, A.N. et al. Inter-domain dynamics in the chaperone SurA and multi-site binding to its outer membrane protein clients. *Nat Commun* **11**, 2155 (2020).
2. Pan, S., Yang, C. & Zhao, X.S. Affinity of Skp to OmpC revealed by single-molecule detection. *Scientific Reports* **10**, 14871 (2020).
3. McMorran, L.M., Bartlett, A.I., Huysmans, G.H., Radford, S.E. & Brockwell, D.J. Dissecting the effects of periplasmic chaperones on the in vitro folding of the outer membrane protein PagP. *J Mol Biol* **425**, 3178-91 (2013).
4. Ebie Tan, A., Burgess, N.K., DeAndrade, D.S., Marold, J.D. & Fleming, K.G. Self-association of unfolded outer membrane proteins. *Macromol Biosci* **10**, 763-7 (2010).
5. Schiffrin, B. et al. Dynamic interplay between the periplasmic chaperone SurA and the BAM complex in outer membrane protein folding. *Commun Biol* **5**, 560 (2022).
6. Marx, D.C. et al. SurA is a cryptically grooved chaperone that expands unfolded outer membrane proteins. *Proc Natl Acad Sci U S A* **117**, 28026-28035 (2020).
7. Chamachi, N. et al. Chaperones Skp and SurA dynamically expand unfolded OmpX and synergistically disassemble oligomeric aggregates. *Proc Natl Acad Sci U S A* **119**, e2118919119 (2022).
8. Bitto, E. & McKay, D.B. Binding of phage-display-selected peptides to the periplasmic chaperone protein SurA mimics binding of unfolded outer membrane proteins. *FEBS Lett* **568**, 94-8 (2004).
9. Humes, J.R. et al. The role of SurA PPIase domains in preventing aggregation of the outer-membrane proteins tOmpA and OmpT. *J Mol Biol* **431**, 1267-1283 (2019).
10. Jia, M. et al. Conformational dynamics of the periplasmic chaperone SurA. *Biochemistry* **59**, 3235-3246 (2020).
11. Soltés, G.R., Schwalm, J., Ricci, D.P. & Silhavy, T.J. The activity of Escherichia coli chaperone SurA is regulated by conformational changes involving a parvulin domain. *J Bacteriol* **198**, 921-9 (2016).
12. Harris, P.D. et al. Multi-parameter photon-by-photon hidden Markov modeling. *Nat Commun* **13**, 1000 (2022).
13. Pirchi, M. et al. Photon-by-Photon Hidden Markov Model Analysis for Microsecond Single-Molecule FRET Kinetics. *J Phys Chem B* **120**, 13065-13075 (2016).

14. Kalinin, S., Valeri, A., Antonik, M., Felekyan, S. & Seidel, C.A. Detection of structural dynamics by FRET: a photon distribution and fluorescence lifetime analysis of systems with multiple states. *J Phys Chem B* **114**, 7983-95 (2010).
15. Santoso, Y., Torella, J.P. & Kapanidis, A.N. Characterizing Single-Molecule FRET Dynamics with Probability Distribution Analysis. *ChemPhysChem* **11**, 2209-2219 (2010).
16. Torella, J.P., Holden, S.J., Santoso, Y., Hohlbein, J. & Kapanidis, A.N. Identifying molecular dynamics in single-molecule FRET experiments with burst variance analysis. *Biophys J* **100**, 1568-77 (2011).
17. Hoffmann, A. et al. Quantifying heterogeneity and conformational dynamics from single molecule FRET of diffusing molecules: recurrence analysis of single particles (RASP). *Phys Chem Chem Phys* **13**, 1857-71 (2011).
18. Lerner, E. et al. Toward dynamic structural biology: Two decades of single-molecule Förster resonance energy transfer. *Science* **359**(2018).
19. Merkley, E.D. et al. Distance restraints from crosslinking mass spectrometry: mining a molecular dynamics simulation database to evaluate lysine-lysine distances. *Protein Sci* **23**, 747-59 (2014).
20. Williamson, M.P. Using chemical shift perturbation to characterise ligand binding. *Progress in Nuclear Magnetic Resonance Spectroscopy* **73**, 1-16 (2013).
21. Ruschak, A.M. & Kay, L.E. Methyl groups as probes of supra-molecular structure, dynamics and function. *J Biomol NMR* **46**, 75-87 (2010).

REVIEWER COMMENTS

Reviewer #1 (Remarks to the Author):

Comments on revision of Dual client binding sites in the ATP-independent chaperone SurA

I thank the authors for discussing the raised points. While some of the distinct points have become more clear, others remain still very elusive and I would like to further inquire, because I think it is important to discuss conceptual ideas transmitted by this manuscript. While I see novelty in the NMR experiments, I still doubt new insights from the smFRET experiments. Especially the interpretation and analysis using a Hidden Markov Model raises significant doubts as detailed below.

I would like to return to the smFRET experiments of the manuscript. I appreciate the discussion also of the earlier publication by Chamachi et al.. Looking at that publication I would like to add that Chamachi et al. looked at timescales from ms to μ s timescale, thus also covering the here presented timescales. Thus the argument of different timescales does not suffice to bring the results together. I am convinced that smFRET experiments can reveal many molecular intricacies that remain to be discovered. However, I am puzzled by the distinct state description of an unfolded polymer in solution and on the strict assignment of two states, which is in contrast to the NMR derived diffusive binding picture.

A) Let's assume we have two distinct states and the interconversion rates are on the order of 1 ms (cf Fig 2d). This would mean that using the Arrhenius equation and a prefactor of 10^9 s^{-1} , the barrier between these states is on the order of $10 k_B T$, which is significant and would be considered in protein folding already a distinct state. So in this case, I am wondering if the authors can describe the expanded state. Can this state be observed by NMR? From the state area in Fig. 2a as well as in the SI figures I would assume it is nearly 40% of the population, this could show up in distinct peaks in NMR. Or is the interpretation that each of the two states is again an ensemble of states?

B) I am more of the impression that mpH2MM biases the analysis towards distinct, maybe 2, states. It would be great if the authors could show that mpH2MM is non-biasing this approach. For instance, there are a number of soluble proteins which showed a broad Eraw distribution in presence of high concentration of denaturants and that are typically described by a continuous sampling of many states. Would this be a benchmark?

In polymer physics it is a well described concept that polymers sample the space available with a mean end-to-end (or label-to-label) distance. But two distinct states have not been described. Along these lines it would be spectacular that OmpX shows this behaviour. However, the BIC analyses show no clear minimum (yes I am aware that this is often hard to achieve, but also sometimes an indication that one observes a continuum of states). The authors also provide BVA

plots, however, how do the BVA plots prior to “recoloring” by the mpH2MM filter look? How does the BVA plot change for different burst length selections? If only very short bursts are chosen, the chance for a dynamic molecule are much lower and the authors should see the blue and red distribution arise then in the Eraw and the BVA plot.

C) While in principle I can see that OmpX has two distinct conformations bound to SurA - especially with the two distinct P1 conformations, looking at the potential mpH2MM bias described before, I am now more puzzled of the strict two state picture. Looking again at the presented FRET efficiency distributions in Fig 2 b and c, the state assigned to be bound (dark red) shows a very distinct shoulder to high FRET efficiencies. This is clearly not arising from shot noise broadening and yet assigned to a non-dynamic state by mpH2MM. Is this an additional state? Maybe the compact state as described in Chamachi et al? If yes, it should be considered in the mpH2MM analysis. Side note: The fraction of the blue area appears much smaller than the red area in Fig 2c. Yet Fig. 2e suggests otherwise. Please explain.

D) I would like to disagree with the authors on their reply here:

“- The earlier FRET study also reported binding of multiple SurA to OmpX. Was this also observed here, or can it be excluded? Could the second distinct state be a second SurA binding and sliding?”
“There is no way to tell from our single-molecule data if multiple SurA molecules are bound to OmpX and this is a possibility (and could contribute to the dynamics and expansion we observe). We have now added to the main text to highlight such a possibility.”

smFRET experiments allow for a titration of SurA and monitoring the fraction of bound OmpX. This would allow to be fitted by a transition function, allowing to see one or two bindings and cooperatively. This is an important point especially when studying the SurA core domain, which is very small and would easily allow for multiple binding events. Please explain.

E) I would like to suggest a visualisation of the dynamics using the normalised fluorescence lifetime vs E_{FRET} (see for instance here: <https://doi.org/10.1063/5.0089134>). In this article it is described that given maybe 3 or more states or a given bound state ensemble the classical interpretation using BVA is limited.

Reviewer #2 (Remarks to the Author):

Although the authors have addressed all of my specific concerns, I still have reservations about the novelty of their work. As I mentioned previously, several previous NMR and smFRET studies have demonstrated the binding of OMPs to SurA and documented the conformational changes that both the chaperone and the substrate undergo. As the authors note in their reply, their work expands on those findings and clearly demonstrate that the core domain of SurA expands unfolded OMPs and that the PPIase domains of SurA modulate substrate expansion. While the results reported in the manuscript add more details to our understanding of SurA function, their significance remains

uncertain. The authors do not fully address the primary question posed in the Abstract: precisely how does SurA recognize and bind a highly diverse group of OMP substrates. As noted in the Introduction, studies published ~20 years ago (refs. 57 and 58) showed that aromatic-rich motifs in OMPs drive their interactions with SurA.

Additionally, the authors propose an autoinhibition model in which SurA predominantly exists in an auto-inhibited core-P1 closed conformation. Previous work by Soltes et al. (ref. 52) showed that the crosslinking of the P1 domain to the core domain leads to OMP assembly defects. Including a form of SurA in which the core and P1 domains are crosslinked in the smFRET experiment shown in Figure 2 would help elucidate the conformational dynamics of OmpX and further support the authors' model.

Reviewer Comments (NCOMMS-24-03929A)

Key: black = verbatim referee comment, blue = our response.

Reviewer #1 (Remarks to the Author):

Comments on revision of Dual client binding sites in the ATP-independent chaperone SurA

I thank the authors for discussing the raised points. While some of the distinct points have become more clear, others remain still very elusive and I would like to further inquire, because I think it is important to discuss conceptual ideas transmitted by this manuscript. While I see novelty in the NMR experiments, I still doubt new insights from the smFRET experiments. Especially the interpretation and analysis using a Hidden Markov Model raises significant doubts as detailed below.

The remaining concerns of Reviewer #1 focus on our mpH²MM analysis of unfolded OmpX and its separation of the broad FRET histogram into two distinct FRET states (see detailed responses below). We now realise that such a detailed mpH²MM analysis adds confusion, does not enable straightforward comparison with the previous literature (e.g. Chamachi et al.¹) and, most importantly, is not required to draw the two conclusions from this section of our paper, i.e. that:

- SurA can expand unfolded OMPs using its core domain alone.
- Substrate expansion is modulated by the presence of the SurA PPIase domains.

We have therefore decided to remove the mpH²MM analysis and instead we show the raw smFRET histograms and burst variance analysis. We hope that Reviewer #1 agrees that these changes do not alter our conclusions, and make the results more easily understandable and accessible to all. A focus of our future work will be to characterise how and why unfolded OmpX undergoes dynamics on such long timescales and how this relates to its ability to be chaperoned by SurA which exhibits dynamics on a similar timescale. Such a new manuscript which will focus entirely on smFRET analysis of unfolded proteins with the space to deal with the analysis of the data in detail for aficionados of the technique.

I would like to return to the smFRET experiments of the manuscript. I appreciate the discussion also of the earlier publication by Chamachi et al. Looking at that publication I would like to add that Chamachi et al. looked at timescales from ms to μ s timescale, thus also covering the here presented timescales. Thus the argument of different timescales does not suffice to bring the results together. I am convinced that smFRET experiments can reveal many molecular intricacies that remain to be discovered. However, I am puzzled by the distinct state description of an unfolded polymer in solution and on the strict assignment of two states, which is in contrast to the NMR derived diffusive binding picture.

We had hoped to make this clear in our previous revisions: the mpH²MM analysis allows us to extract two FRET states of unfolded OmpX on the ms timescale. These are not single structures, but an ensemble of many conformations in rapid exchange within each state's energy well (sub ms). The raw smFRET distribution is broad (and fully consistent with similar experiments by Chamachi et al.), as we now show (see below). In turn, these conformational ensembles

(represented by two FRET states) are in a slower (ms) exchange with each other.

A) Let's assume we have two distinct states and the interconversion rates are on the order of 1 ms (cf Fig 2d). This would mean that using the Arrhenius equation and a prefactor of 10^9 s^{-1} , the barrier between these states is on the order of $10 k_B T$, which is significant and would be considered in protein folding already a distinct state. So in this case, I am wondering if the authors can describe the expanded state. Can this state be observed by NMR? From the state area in Fig. 2a as well as in the SI figures I would assume it is nearly 40% of the population, this could show up in distinct peaks in NMR. Or is the interpretation that each of the two states is again an ensemble of states?

It is vital to note here that unfolded OmpX is not a Florey polymer, nor it is an IDP. Instead, it is an unfolded, collapsed chain, poised to fold once it encounters a bilayer. Non-random-coil chain behaviour of unfolded OMPs in low amounts of denaturant (as used here – 0.1 M urea) is well-established in the field (see Krainer et al., *Biophys J*, 2019, Marx et al., *PNAS*, 2020, Chamachi et al., *PNAS*, 2022)¹⁻³. Moreover, slow (\geq ms) timescale dynamics are hypothesised to be limited by the formation of long-range tertiary interactions in the unfolded-state ensemble (Krainer et al., 2019)². We are thus not at all surprised that unfolded OmpX populates more and less collapsed states that exchange on a ms timescale.

Characterization of individual conformers within heterogeneous unfolded protein ensembles by NMR is an extremely challenging task. While for folded proteins, two conformations would usually result in two sets of peaks in a corresponding NMR spectrum (if in slow exchange), conformations in rapid exchange are very hard to distinguish and their characterization is a project in itself, requiring acquisition of a large set of experimental data (e.g., PRE, relaxation dispersion, R1/R2, RDCs, etc)⁴. This is a large undertaking and will be the focus of our future work.

B) I am more of the impression that mpH²MM biases the analysis towards distinct, maybe 2, states. It would be great if the authors could show that mpH²MM is non-biasing this approach. For instance, there are a number of soluble proteins which showed a broad Eraw distribution in presence of high concentration of denaturants and that are typically described by a continuous sampling of many states. Would this be a benchmark?

We agree that unfolded OmpX will populate many different conformations, but there are two that can be separated and resolved based on their different FRET values using mpH²MM. We have now removed the mpH²MM analysis from the text. However, based on the referee's suggestion we have obtained smFRET data on unfolded OmpX in 4 M urea, which has very different E_{max} than unfolded OmpX in 0.1 M urea (our standard FRET conditions) and does not exhibit dynamics in the ms timescale (as demonstrated using BVA (Fig. S1)), as expected for a highly denatured chain. So, while we completely agree with Reviewer #1 on how a simple unfolded polypeptide should behave, this is not what we see for OmpX in 0.1 M urea, and our data (and that of Chamachi et al.¹) is clear evidence of this non-random behaviour.

Note, since submission we have newly built our smFRET setup, as now detailed in the Supplementary Methods. As we are including the new data for OmpX in 4 M urea, we have re-acquired all the smFRET data with OmpX/SurA at lower urea concentration using our new

instrumentation so we can directly compare all the raw FRET histograms. This has not changed the results, or our conclusions, but needed to be done to allow this comparison.

In polymer physics it is a well described concept that polymers sample the space available with a mean end-to-end (or label-to-label) distance. But two distinct states have not been described. Along these lines it would be spectacular that OmpX shows this behaviour. However, the BIC analyses show no clear minimum (yes I am aware that this is often hard to achieve, but also sometimes an indication that one observes a continuum of states). The authors also provide BVA plots, however, how do the BVA plots prior to “recoloring” by the mpH2MM filter look? How does the BVA plot change for different burst length selections? If only very short bursts are chosen, the chance for a dynamic molecule are much lower and the authors should see the blue and red distribution arise then in the E_{raw} and the BVA plot.

BVA plots and smFRET histograms of the full dataset are now included in a new Fig. 2. Importantly, our main conclusions from the smFRET data (that the expansion of unfolded OmpX does not require all three domains in SurA-WT and that the extent of expansion is modulated by the PPlase domains) can be seen in the raw smFRET data (E_{raw} values). As we have decided to remove the mpH2MM analysis we have not analysed the data further here with respect to the burst length.

C) While in principle I can see that OmpX has two distinct conformations bound to SurA - especially with the two distinct P1 conformations, looking at the potential mpH2MM bias described before, I am now more puzzled of the strict two state picture. Looking again at the presented FRET efficiency distributions in Fig 2 b and c, the state assigned to be bound (dark red) shows a very distinct shoulder to high FRET efficiencies. This is clearly not arising from shot noise broadening and yet assigned to a non-dynamic state by mpH2MM. Is this an additional state? Maybe the compact state as described in Chamachi et al? If yes, it should be considered in the mpH2MM analysis. Side note: The fraction of the blue area appears much smaller than the red area in Fig 2c. Yet Fig. 2e suggests otherwise. Please explain.

Given the complexity of the mpH2MM analysis that is not required for the conclusions of our manuscript, and the confusion regarding the number of distinct FRET states versus protein conformations, we have decided to remove the mpH2MM analysis from our manuscript and show instead the raw smFRET data. Note that the results have not changed. This removes all ambiguity, increases clarity, and allows direct comparison with Chamachi et al.¹⁾, with which our data concur (any differences can be attributed to different dyes and differences in solution conditions and the manner in which unfolded OmpX was prepared (ours was never folded in detergent whilst Chamachi et al. commenced with LDAO folded native OmpX before diluting out detergent and adding 6M GdmCl (Fig. 1A of Chamachi et al.)).

D) I would like to disagree with the authors on their reply here: “- The earlier FRET study also reported binding of multiple SurA to OmpX. Was this also observed here, or can it be excluded? Could the second distinct state be a second SurA binding and sliding?” “There is no way to tell from our single-molecule data if multiple SurA molecules are bound to OmpX and this is a possibility (and could contribute to the dynamics and expansion we observe). We have now added to the main text to highlight such a possibility.”

smFRET experiments allow for a titration of SurA and monitoring the fraction of bound OmpX. This would allow to be fitted by a transition function, allowing to see one or two bindings and cooperatively. This is an important point especially when studying the SurA core domain, which is very small and would easily allow for multiple binding events. Please explain.

Rather than using smFRET for analysis of the cooperativity of binding, we have determined the Hill coefficient for the binding of SurA-WT and SurA-core to unfolded OmpX using microscale thermophoresis (MST), as is used for tBamA:OmpX binding in Fig. S7 of our current submitted manuscript. The results show that the Hill coefficient of SurA-WT and SurA-core binding to unfolded OmpX under the conditions of our experiments are very similar. Hence, whilst we cannot determine how many SurA molecules bind to unfolded OmpX, we can be confident that cooperativity of binding in the two samples is the same. We include these results as a new Supplementary Fig. S2.

E) I would like to suggest a visualisation of the dynamics using the normalised fluorescence lifetime vs E_FRET (see for instance here: <https://doi.org/10.1063/5.0089134>). In this article it is described that given maybe 3 or more states or a given bound state ensemble the classical interpretation using BVA is limited.

We have now included corrected FRET efficiency vs normalised fluorescence lifetime plots in a new Fig. S1. The results show that for apo-OmpX in 4 M urea, dynamics can be observed in the lifetime analysis (\leq ms) but do not manifest in the BVA analysis (= ms). By contrast in 0.1 M urea, dynamics are observed on both timescales. New text to discuss this has now been included in the results section.

Reviewer #2 (Remarks to the Author):

Although the authors have addressed all of my specific concerns, I still have reservations about the novelty of their work. As I mentioned previously, several previous NMR and smFRET studies have demonstrated the binding of OMPs to SurA and documented the conformational changes that both the chaperone and the substrate undergo. As the authors note in their reply, their work expands on those findings and clearly demonstrate that the core domain of SurA expands unfolded OMPs and that the PPIase domains of SurA modulate substrate expansion. While the results reported in the manuscript add more details to our understanding of SurA function, their significance remains uncertain. The authors do not fully address the primary question posed in the Abstract: precisely how does SurA recognize and bind a highly diverse group of OMP substrates. As noted in the Introduction, studies published ~20 years ago (refs. 57 and 58) showed that aromatic-rich motifs in OMPs drive their interactions with SurA.

Additionally, the authors propose an autoinhibition model in which SurA predominantly exists in an auto-inhibited core-P1 closed conformation. Previous work by Soltes et al. (ref. 52) showed that the crosslinking of the P1 domain to the core domain leads to OMP assembly defects. Including a form of SurA in which the core and P1 domains are crosslinked in the smFRET experiment shown in Figure 2 would help elucidate the conformational dynamics of OmpX and further support the authors' model.

We have not attempted this experiment, although we would like to do so as part of future work. It will not be straightforward to perform, or analyse, however. Previous studies have shown that cross-linking P1 to Core substantially reduces unfolded OMP binding (Marx *et al.*, PNAS, 2020³) which will make complex formation difficult to achieve in any significant amount, and interpretation of any results difficult to interpret without controls and a detailed analysis of how the cross-linked protein behaves.

References

1. Chamachi, N. et al. Chaperones Skp and SurA dynamically expand unfolded OmpX and synergistically disassemble oligomeric aggregates. *Proc Natl Acad Sci U S A* **119**, e2118919119 (2022).
2. Krainer, G. et al. Slow Interconversion in a Heterogeneous Unfolded-State Ensemble of Outer-Membrane Phospholipase A. *Biophys J* **113**, 1280-1289 (2017).
3. Marx, D.C. et al. SurA is a cryptically grooved chaperone that expands unfolded outer membrane proteins. *Proc Natl Acad Sci U S A* **117**, 28026-28035 (2020).
4. Camacho-Zarco, A.R. et al. NMR Provides Unique Insight into the Functional Dynamics and Interactions of Intrinsically Disordered Proteins. *Chemical Reviews* **122**, 9331-9356 (2022).

REVIEWERS' COMMENTS

Reviewer #1 (Remarks to the Author):

Schiffrin, Crossley and coworkers studied in this manuscript the binding of several OMPs and b-strand peptides to SurA using NMR and smFRET. I appreciate the authors edits on the manuscript. I see all my comments addressed.

In detail, I see that the authors have removed the HMM data analysis from the manuscript. While it is challenging to remove such an analysis, I am convinced the way how the smFRET data is now presented it is most meaningful and long-lasting. It agrees with earlier studies, but also reveals new features like the binding to the SurA core domain. The data also shows that the core domain binds OmpX differently than full-length SurA. I also appreciate the kinetics and FRET-lines analysis added to the supplementary information.

I do not have any further comments. Congrats on a very nice manuscript!